# Deep Learning-Based Recognition and Classification of Soiled Photovoltaic Modules Using HALCON Software for Solar Cleaning Robots

**DOI:** 10.3390/s25051295

**Published:** 2025-02-20

**Authors:** Shoaib Ahmed, Haroon Rashid, Zakria Qadir, Qudratullah Tayyab, Tomonobu Senjyu, M. H. Elkholy

**Affiliations:** 1Department of Electrical and Electronic Engineering, Faculty of Engineering, University of Ryukyus, 1 Subaru, Nishihara 903-0213, Nakagami, Okinawa, Japan; qudrat.tayyab@gmail.com (Q.T.); a985542@yahoo.co.jp (T.S.); k228677@cs.u-ryukyu.ac.jp (M.H.E.); 2Institut de Recherche Dupuy de Lôme (UMR CNRS 6027), University of Brest, 29238 Brest, France; haroon.rashid@univ-brest.fr; 3Sydney International School of Technology and Commerce, Sydney, NSW 2000, Australia

**Keywords:** solar power, soiling, computer vision, deep learning, solar module cleaning system

## Abstract

The global installation capacity of solar photovoltaic (PV) systems is exponentially increasing. However, the accumulation of soil and debris on solar panels significantly reduces their efficiency, necessitating frequent cleaning to maintain optimal energy output. This study presents a deep learning-based approach for the recognition and classification of soiled PV images, aimed at enhancing the capabilities of solar cleaning robots through the HALCON software framework. Using EANN and CNN architecture along with advanced image processing techniques, the proposed system achieves precise detection and classification of soiling patterns. The HALCON framework facilitates image acquisition, preprocessing, segmentation, and deployment of trained models for robotic control. The trained models demonstrate exceptional accuracy, with the EANN and CNN achieving classification precision of 99.87% and 99.91%, respectively. Experimental results highlight the system’s potential to improve automation of cleaning strategies, reduce unnecessary cleaning cycles, and enhance the overall performance of solar panels. This research underscores the transformative role of intelligent visual analysis in optimizing maintenance practices for renewable energy applications.

## 1. Introduction

Photovoltaic (PV) technology is pivotal in the global transition to renewable energy, offering a clean and sustainable means of electricity generation [1,2]. As the installation of PV systems increases, maintaining their efficiency has become a critical challenge [3]. One of the most significant factors affecting the performance of PV modules is soiling, which involves the accumulation of dust, dirt, bird droppings, pollen, and other particulates on the surface of solar panels [4,5]. This layer of contaminants can obstruct sunlight, leading to substantial reductions in energy output [6,7,8]. Early studies quantified the impact of soiling on PV performance, revealing that energy losses could be as high as 50% in areas with heavy dust accumulation [9]. It normally varies with the tropical conditions of the country. The countries most affected by soiling are shown in Figure 1 [10].

However, soiling comprises multiple factors [11], including the strength of the radiated energy, the light direction of the module relative to the sun, the inclination, the geographic location, the ambient temperature, surface shadows, and dirt, which are taken into consideration [12,13]. The interlinks between the sources of soiling on solar modules are shown in Figure 2.

The accumulation of foreign particles on the panel surface not only reduces the amount of solar radiation that can enter but also raises the temperature of the cells, resulting in a twofold effect [14]. In most cases, pollution-related losses are estimated to be between 3 and 8 percent of the total economic output; however, these findings underscored the necessity of effective cleaning and maintenance strategies, particularly for large-scale solar installations.

Initial methods for addressing soiling were largely manual or semi-automated, and although automated systems emerged, they typically lacked the precision needed for optimizing cleaning processes, as they operated on fixed schedules without considering the actual level of soiling on individual panels [15]. Recent studies have explored machine learning-based approaches for soiling detection in PV systems [16,17,18]. One study applied visible spectrum imaging and various classifiers, including convolutional neural networks (CNNs), support vector machines (SVMs), random forests (RFs), and k-nearest neighbors (kNN), to identify soiling types such as dust and bird droppings [19]. The study emphasized image preprocessing techniques to enhance feature extraction. While different models demonstrated varying performance, CNNs showed strong adaptability during real-world testing. These findings align with ongoing efforts to integrate deep learning (DL) into PV maintenance, reinforcing the significance of automated approaches for improving operational efficiency. A study utilized various image features to develop a machine learning regression model, enabling accurate estimation of soiling loss levels [20]. The model was trained and tested with real-world PV performance data and RAW panel images collected over several months, covering soiling loss levels of up to 28%. The dataset consisted of 479 RAW images with 21 distinct soiling loss levels, captured under varying camera settings.

Recent research has explored the use of deep learning techniques for detecting defects in photovoltaic panels using aerial and electroluminescence imaging [21]. The study utilized two distinct datasets to capitalize on the strengths of each imaging method, with aerial images providing insights into surface-level defects and electroluminescence images revealing internal faults. To improve detection accuracy, different deep learning models were applied, including DenseNet121 and MobileNetV3 for aerial image classification, while a customized ELFaultNet architecture and EfficientNetV2B2 were used for electroluminescence image analysis. The models achieved high classification accuracy, with DenseNet121 reaching 93.75 percent and MobileNetV3 achieving 93.26 percent, while ELFaultNet and EfficientNetV2B2 recorded 91.62 percent and 81.36 percent, respectively. These results highlight the potential of deep learning models for enhancing photovoltaic panel monitoring, facilitating early defect identification and improving maintenance efficiency. A study introduced a Vision Transformer (ViT)-based ANN model for automated PV fault classification, leveraging image preprocessing, data augmentation, and transformer-based deep learning for enhanced precision [22]. The approach applied an unsharp mask technique to refine image edges and data augmentation to mitigate class imbalance issues. The model was evaluated using the Infrared Solar Modules dataset, classifying eleven PV fault types along with a no-anomaly category. The proposed method achieved 98.23 percent accuracy for binary classification, 96.19 percent accuracy for identifying eleven fault types, and 95.55 percent accuracy when incorporating the no-anomaly category. These results demonstrate the effectiveness of transformer-based architectures in enhancing PV fault detection accuracy, offering a promising approach for early anomaly identification and improved maintenance strategies. Rahmouni et al. [23] proposed a novel approach for diagnosing PV cell degradation using drone-acquired thermal imagery. Their work leverages the YOLO (You Only Look Once) deep learning model to detect abnormal cells based on temperature anomalies. By using thermal image datasets, the study achieved high accuracy and real-time detection capabilities, addressing the challenges of diagnosing inaccessible PV systems. This approach demonstrates the potential of combining drone technology with advanced image analysis tools for efficient PV fault detection.

As deep learning models evolved, compact and efficient architectures, such as Mobile Net and Efficient Net [24], were developed to overcome the computational challenges of deploying CNNs in real-time systems. These architectures, introduced in [25], respectively, maintained high accuracy while reducing computational demands, making them suitable for edge devices and autonomous cleaning robots. Simultaneously, machine vision software like HALCON emerged as a vital tool in enabling real-time image processing and deep learning-based classification. HALCON’s flexibility and support for CNN integration made it an ideal platform for implementing automated soiling detection systems. Liebig and Bretthauer [26] demonstrated HALCON’s effectiveness in industrial applications, which laid the groundwork for its use in PV module classification and reorganization. Berghout et al. [27] conducted a comprehensive review of machine learning-based condition monitoring techniques for PV systems. Their study categorizes various machine learning paradigms, including traditional, hybrid, and ensemble frameworks, and explores their application in failure detection, diagnosis, and prediction. The review also emphasizes the growing importance of deep learning architectures, such as adversarial networks and transfer learning, in enhancing prediction accuracy and robustness. Furthermore, the work highlights the availability of open-source datasets, which are critical for developing and validating data-driven fault detection systems.

The combination of CNNs and HALCON software has also been integrated into autonomous solar cleaning robots, further improving the efficiency of cleaning operations. The authors of [28] demonstrated the effectiveness of these robots in using CNN-powered real-time soiling detection systems, allowing for them to clean only the most affected areas, thereby conserving water and energy. This shift towards intelligent, selective cleaning represents a significant improvement over earlier, less efficient methods. Recent advancements, as noted by Chowdhury et al. [29], focus on enhancing the generalizability of these models by training them on more diverse datasets, ensuring their robustness in different environmental conditions. Additionally, there is growing interest in developing fully autonomous systems capable of real-time decision-making and operation, which will further enhance the sustainability and scalability of PV maintenance.

### Research Gaps and Contributions

The efficiency of PV systems is significantly impacted by soiling, which includes the accumulation of dust, dirt, and other environmental contaminants on solar panels. While automated cleaning systems have been introduced, most of these operate on fixed schedules rather than real-time soiling detection. This results in inefficient maintenance, either excessive cleaning that wastes water and energy or delayed cleaning that leads to energy losses. An effective solution should be capable of adaptive, intelligent cleaning, triggered only when necessary, based on real-time classification of soiling levels. Current deep learning-based classification models for soiling detection are computationally demanding, making them difficult to deploy on autonomous robots with limited processing capabilities. Many existing approaches use standard CNNs, which, while accurate, require substantial computational power. This poses a significant challenge for real-time implementation in embedded systems, emphasizing the need for lightweight yet accurate deep learning architectures that can function efficiently in resource-constrained environments.

Moreover, the potential of machine vision software, such as HALCON, in PV maintenance remains underexplored. HALCON provides powerful image acquisition, segmentation, and classification tools, yet its integration with deep learning models for real-time PV soiling detection has not been thoroughly investigated. Combining deep learning with HALCON’s robust image processing capabilities could enhance efficiency and reliability in automated solar cleaning systems. Additionally, there is a lack of comparative studies evaluating different deep learning architectures for PV soiling detection. While CNNs have been widely studied, the effectiveness of an EANN compared to compact CNNs in PV applications has not been systematically analyzed. A comprehensive evaluation of these models is necessary to determine the optimal balance between accuracy, inference speed, and computational efficiency for real-world deployment.

To address these limitations, this study introduces a novel approach that enhances automation in PV module cleaning by integrating deep learning-based classification with HALCON software. The key contributions of this research are as follows:Implementation of EANN and compact CNN architectures for accurate and efficient classification of soiled PV panels, enabling real-time detection of soiling patterns. This approach allows for targeted cleaning, optimizing energy and water usage while reducing unnecessary maintenance cycles.Utilization of HALCON’s machine vision capabilities for image preprocessing, segmentation, and classification, improving efficiency in PV maintenance.Assessment of EANN and CNN architectures in terms of accuracy, inference time, and computational efficiency for real-time deployment, demonstrating superior classification performance with significantly reduced processing time, making them suitable for autonomous PV maintenance applications.Enhancement of solar farm sustainability through improved operational efficiency, optimized resource utilization, and prolonged PV panel lifespan.

Section 2 presents the data collection and analyses. Section 3 describes the EANN datasets, models, and results evaluation. Section 4 covers the CNN datasets, models, and results evaluation. Section 5 describes the electrical and 3D mechanical design of the solar module cleaning system, and Section 6 describes the conclusions.

## 2. Data Collection and Analysis

### 2.1. Data Collection

The data collected from the internet comprises 3497 pictures of solar panels [30]. In the test setup stated in [26], two identical solar panels were positioned side by side, with an RGB camera capturing images at 5 s intervals. The first panel (near the camera) was designated for soiling tests, while the second served as a reference. The generated electricity was recorded alongside the images to analyze the impact of soiling on power output. To ensure a well-structured dataset, a systematic approach for image labeling, data augmentation, and class balancing was implemented. The dataset was categorized based on soiling intensity and defect type, ensuring accurate classification for supervised learning. Labels were assigned through manual expert annotation, supported by automated preprocessing techniques that analyzed color, texture, and shape variations to refine classification accuracy. Multiple augmentation techniques were applied to improve model generalization and reduce overfitting, including random rotations, brightness adjustments, horizontal flipping, contrast normalization, and Gaussian noise addition. These augmentations increased dataset diversity while preserving essential image features for classification. A hybrid strategy was implemented to address class imbalance, combining oversampling of underrepresented classes with augmentation-based sample balancing. Additionally, class-weighted loss functions were incorporated during training to ensure equal learning across all categories and prevent bias toward dominant classes.

In comparison to the reference panel, the percentage of power loss due to soiling is reported. The goal of the data-gathering technique was to capture various types of soiling and their consequences on PV panels. The data collection strategy was designed to capture different types of soiling and their effects on PV panels. Under natural environmental conditions, we exposed the panel to several types of soiling in terms of color (red, brown, and grey), particle size (sand, dust, and talcum powder), and thickness. Some of the thick spots correspond to significant power losses of up to 90%. The dataset was augmented with considerable differences in soiling owing to both experimental and natural means (e.g., dust with varying thickness, blob sizes, and patches) (e.g., wind and precipitation as shown in below Figure 3, Figure 4, Figure 5 and Figure 6).

### 2.2. Data Processing

In the proposed approach, images of solar panels undergo a multi-step preprocessing pipeline in HALCON software before being fed into the deep learning model. The purpose of these preprocessing steps is to enhance image quality, normalize variations in environmental conditions, and improve the robustness of the classification model. The following steps were applied to the dataset to ensure consistent and reliable inputs for model training:Image Resizing: All images were resized to 224 × 224 pixels to standardize input dimensions, ensuring uniformity across the dataset. This resizing allows for the model to efficiently learn feature representations without scale distortions.Color Normalization: Given that images were collected under various lighting conditions, RGB normalization was applied to adjust pixel intensity distributions. This step mitigates inconsistencies due to varying brightness and shadows, ensuring that color-based features remain consistent.Noise Reduction: To enhance the clarity of key features such as dirt accumulation and cracks, a Gaussian filter was applied. This helps smooth out unwanted noise while preserving edge information, ensuring that small defects remain distinguishable.Histogram Equalization: This technique was implemented to improve image contrast, enhancing visibility of important details. Uneven lighting or shadows can create inconsistencies in pixel intensity, and histogram equalization balances the brightness levels, making the dataset more uniform.Data Augmentation: To improve the generalization capability of the model and prevent overfitting, the following data augmentation techniques were applied:
-Random Rotations (±15 degrees): Helps the model learn rotational invariance and handle different panel orientations.-Brightness Adjustments (±20%): Accounts for illumination differences due to environmental conditions.-Horizontal Flipping (50% probability): Improves robustness by introducing variations in soiling patterns.-Contrast Normalization: Ensures that variations in brightness do not impact feature extraction.Pixel Value Normalization: All pixel intensities were rescaled to the [0,1] range. Normalizing pixel values helps prevent large variations in gradient magnitudes, leading to better optimization convergence and improved training stability.Image Labeling: The dataset was annotated based on soiling intensity and defect type, ensuring precise classification. Labels were assigned using a combination of manual expert annotation and automated preprocessing techniques, where key features such as color, texture, and shape were analyzed to improve annotation accuracy.Class Imbalance Handling: To address class imbalance, a hybrid approach was employed, combining oversampling of underrepresented classes and augmentation-based sample balancing. Additionally, class-weighted loss functions were integrated into the training process to ensure equal representation of all classes and prevent bias toward majority classes.

These preprocessing techniques ensure that the dataset is well prepared for deep learning, enhancing feature extraction capabilities and improving model performance. The refined and augmented dataset provides a robust foundation for training a highly accurate classification model.

### 2.3. RGB Images of Solar Panel and Its 3D Plot

An RGB (red, green, blue) image displays the solar panel in normal color format, showing visible defects and dirt that affect the panel. HALCON processes these images by applying filters, edge detection, and other computer vision algorithms to identify features of interest, like soiling patterns. An RGB image is a three-dimensional byte array in which each pixel’s color value is explicitly stored [31]. The RGB image of the panel depicted three color channels and a width and height. RGB images are often used to store scanned photographs. The color data are recorded in three portions of the image’s third dimension. Solar panel RGB images are shown in Figure 7b, Figure 8b, Figure 9b, and Figure 10b. The color channels, color bands, and color layers are the terms used to describe these areas. The amount of red in the image is represented by one channel (the red channel shows the clean region of the panel), the amount of green in the image is represented by one channel (the green channel shows the dirty region of the panel), and the amount of blue in the image is represented by one channel (the blue channel shows the edges of the panel).

Three-dimensional plots are helpful for analyzing surface irregularities, determining the height or depth of soiling, and detecting damage like cracks. In the 3D plots shown in Figure 7c, Figure 8c, Figure 9c, and Figure 10c, the green spikes show dirty regions while the red spikes show clean regions of the panel.

### 2.4. RGB Histogram of Solar Panels

RGB histogram of solar panels using HALCON; the process involves analyzing the red, green, and blue color channels of the RGB image of the solar panel [32,33,34]. The histogram shows the distribution of pixel intensity values for each color channel (R, G, B), which is useful in assessing soiling levels, identifying defects, or understanding the overall color composition of the image. HALCON provides the functionality to separate the RGB image into its three-color channels [35]. This is achieved by splitting the image into its red, green, and blue components. After separating the color channels, histograms for each are to be computed using HALCON’s histogram function. This gives the distribution of pixel intensity values for red, green, and blue separately, as shown in Figure 7d, Figure 8d, Figure 9d, and Figure 10d. The histograms show the frequency of pixel intensity values (ranging from 0 to 255) for the red, green, and blue components, allowing for observation of how much each channel contributes to the overall image.

This information is also available in text format in the Statistics section of the display table. The histogram’s peak, i.e., the grey value that appears the most frequently, is likewise shown in the statistics (see below). The peak value is given as a value range in the Statistics Analysis of the RGB histogram shown in Table 1. To generate a meaningful histogram, the range of input values is divided into quantization steps, and as a result, the histogram’s “peak value” can accurately reflect the entire range of input values.

### 2.5. Histogram of Malfunctioned Region of Solar Panel

A histogram of a malfunctioned region of solar panels refers to the graphical representation of pixel intensity values from an image of the region where the malfunction (such as cracks, defects, or dirt accumulation) is occurring. A histogram is a graph or plot that shows the frequency of pixels in an RGB image. The source image plot *X*-axis represented as an image has pixel values (ranging from 0 to 255). The channel index is 2, for which the histogram is calculated. This is our range, as represented by the size of the histogram. It is usually [0, 256]. In Figure 11, Figure 12 and Figure 13 below, the histogram of the malfunctioned region of the solar module is shown as compared to all the histograms. The highly malfunctioned region shows high peaks and soiling percentage, and the statistical analysis is shown in Table 2.

### 2.6. Heat Map of Different Solar Images

The heat map of a dataset sample displayed below demonstrates the region of the solar panel having a significant impact on inference on a specific class during mode training, as shown in Figure 14, Figure 15, Figure 16 and Figure 17.

## 3. Deep Learning-Based Models and Results Evaluation

### 3.1. Image Classification Using Enhanced Artificial Neural Network

The EANN is a deep learning architecture designed to handle complex classification problems by incorporating multiple hidden layers as shown in Figure 18. The multi-layer structure allows for the model to learn hierarchical representations, making it well suited for distinguishing between different soiling patterns on PV modules. The MVTec HALCON framework supports the implementation of an EANN, providing robust tools for deep learning-based classification tasks. However, increasing the number of hidden layers and network complexity requires more computational resources, leading to higher memory consumption and longer processing times [36,37].

In an EANN, the classification process relies on computing the weighted sum of inputs at each neuron, followed by the application of a nonlinear activation function to introduce learning capabilities beyond simple linear decision boundaries. The mathematical formulation of the EANN model is given as follows:
(1)$$a_{j}=\sum_{i=1}^{n}(W_{i,j}X_{i}+b_{j})$$
where $a_{j}$ represents the aggregated weighted sum of inputs for neuron j, and $X_{i}$ denotes the input features such as pixel intensities, color distributions, or extracted feature maps from the PV images. The term $W_{i,j}$ represents the weight parameters that determine the strength of the connection between input $X_{i}\;\mathrm{to}\;\mathrm{neuron}\;j$, while $b_{j}$ is the bias term that helps shift the activation function to improve learning performance. The summation runs over all input features, ensuring that every input contributes to the neuron’s activation.

After computing the weighted sum, a nonlinear activation function is applied to introduce learning capabilities that allow for the network to model complex relationships between inputs. This is expressed as
(2)$$Y_{j}=f(a_{j})=f(\sum_{i=1}^{n}W_{i,j}X_{i}+b_{j})$$
where $Y_{j}$ is the final output of neuron j after activation, and $f\left(\cdot \right)$ represents the chosen activation function. Activation functions play a crucial role in enabling the network to capture nonlinear patterns in the data. Common activation functions include ReLU, which is computationally efficient and helps prevent the vanishing gradient problem; Sigmoid, which maps inputs to a probability-like range (0,1); and Tanh, which scales inputs within the range (−1,1) and is useful for handling gradients in deeper networks.

The activation function significantly influences the gradient flow during backpropagation, affecting the model’s ability to learn complex patterns. A well-chosen activation function ensures that the network generalizes effectively to new data, reducing the risk of overfitting or underfitting. By leveraging multiple hidden layers and nonlinear activations, an EANN can learn intricate representations of soiling patterns, leading to more accurate classification.

In the context of PV module soiling classification, an EANN offers several advantages: The hierarchical structure enables progressive feature extraction, allowing for the network to distinguish between different types of soiling such as dust, bird droppings, and snow. Additionally, the multiple hidden layers facilitate adaptive learning, where the network adjusts its weight parameters over successive training iterations to optimize classification accuracy. This adaptability is particularly useful for handling varying environmental conditions, ensuring that the model remains robust across different datasets.

Despite its advantages, an EANN requires careful balancing between network complexity and computational efficiency. Increasing the number of layers improves learning capacity but also raises memory consumption and inference time, which can be a limiting factor for real-time applications. Therefore, selecting the appropriate network depth, activation functions, and training parameters is crucial for achieving high performance without excessive resource demands.

### 3.2. Flowchart of Enhanced Artificial Neural Network Model

Figure 19 shows a flowchart of the image classification process, which is carried out using MvTec’s HALCON deep learning tool. Python is the programming language used in the software, with HALCON 24.11 Progress-Steady as the vision library. The systems are launched by acquiring images of the solar panel utilizing an overhead camera on the robot, as shown in the flowchart. Filters are used to process the image. After that, the model is trained using images. HALCON offers pre-trained models such as EANN, RESnet-50, and Mobile Net V2. Then, for testing, unseen panel images are used. After the output has been classified into the appropriate type of malfunctioning solar panel, the procedure ends.

The flowchart begins with an image of a solar panel captured live by a robot’s digital camera. The EANN is used to apply this image. The trained model classifies the image as either clean or malfunctioning. If the solar panel is anomalous with dirt, the robot starts; if the solar panel is clean, the robot stops.

### 3.3. Experimental Dataset

The solar panel digital image dataset comprises 45,754 solar panel images. However, we only used 3497 images (224 × 224) of them. Image label statistics are shown in Figure 20.

The training dataset is made up of about 71.98% of the data, the test set is made up of about 11.98% of the data, and the validation set is 16.04%. The complete dataset distribution is shown in Table 3 with no images.

### 3.4. Experimental Configuration

The deep learning model was trained using the MVTec HALCON deep learning framework on a system with an Intel(R) Core i7-8550U CPU (1.80 GHz) and 8 GB RAM. A set of carefully chosen hyperparameters was employed to balance model accuracy, training efficiency, and computational feasibility. The selected hyperparameters are as follows:Input Image Size (224 × 224 pixels): To maintain uniformity across the dataset and ensure compatibility with deep learning architectures, all images were resized to 224 × 224 pixels. This standardization prevents discrepancies in resolution and maintains feature consistency across different training samples.Batch Size (33): The batch size determines the number of images processed before an update to the model’s weights is made. A batch size of 33 was chosen as an optimal trade-off between training stability and memory constraints. A smaller batch size would have led to unstable gradient updates, while a larger batch size would have required excessive memory.Epochs (25): The model was trained for 25 epochs, which was determined based on empirical validation. The number of epochs was selected to allow for sufficient learning while preventing overfitting. Beyond 25 epochs, the validation loss did not show significant improvement.Learning Rate (0.001): The learning rate controls how much the model’s weights are adjusted during training. A learning rate of 0.001 was chosen to ensure gradual convergence without large fluctuations in weight updates. Higher learning rates caused instability in training, while lower values resulted in slow convergence.Momentum (0.9): Momentum is used to smooth out gradient updates and accelerate convergence. A momentum value of 0.9 helps prevent oscillations and speeds up learning in the early training stages. This was found to be particularly effective in stabilizing optimization when using stochastic gradient descent (SGD).Weight Decay (0.0001): To prevent overfitting, L2 regularization (weight decay) of 0.0001 was applied. This regularization term discourages large weight values, improving the generalization ability of the model.Activation Function (ReLU): The Rectified Linear Unit (ReLU) activation function was selected for its effectiveness in avoiding the vanishing gradient problem. It introduces nonlinearity while maintaining computational efficiency.Optimizer (Adam): The Adam optimizer was used due to its adaptive learning rate adjustment mechanism, which helps balance speed and stability in training. Compared to SGD, Adam demonstrated faster convergence and better generalization on the validation set.Loss Function: The model was trained using categorical cross-entropy loss, as it is well suited for multi-class classification problems. This loss function calculates the divergence between predicted probabilities and actual labels, allowing for the model to optimize its predictions effectively.

The selection of these hyperparameters was based on multiple experimental trials, and adjustments were made to fine-tune the training process. The chosen values provide an optimal balance between computational efficiency and classification accuracy, ensuring that the model can be deployed effectively in real-world applications.

### 3.5. Model Results and Evaluation

In this section, we evaluate the trained deep learning models’ validity. The overall evaluation is shown in Table 4.

Accuracy: The ratio between the number of correct predictions and the total number of predictions in our model and the number of corrected predicted images were 418 and 419, so for overall accuracy, we found 99.76% for the EANN model.
(3)$$A=\frac{\left(TP+TN\right)}{\left(TP+TN+FP+FN\right)}=\frac{418}{419}\times 100=99.76\%$$
where TP represents “true positives”, TN represents “true negatives”, FP represents “false positives”, and FN represents “false negatives”.Inference time: Inference time refers to the time taken by a trained model (such as an ANN) to process a single input (like an image) and generate an output (like a classification result). In the context of solar panel image classification using enhanced an ANN in HALCON, inference time is an important metric for evaluating the efficiency of the model in real-time or near-real-time applications. The inference time is 15.73.Top-1 error: The proportion of images for which the predicted class is incorrect. Our model’s top 1 error is 0.24%.F1 Score: The precision and recall harmonic mean is called the F1 score for each class of the dataset shown in Table 5, and the overall F1 score of the models is 99.85.Precision: All accurately predicted positives as a proportion of all predicted positives. The overall model means precision is 99.87%. Individual dataset precision is shown in Table 5.
(4)$$P=\frac{TP}{\left(TP+FP\right)}$$Recall: All accurately predicted positives as a proportion of all actual positives. The recall is a metric that evaluates how well a model can detect positive samples. The overall predicted recall percentage of the model is 99.83%.
(5)$$R=\frac{TP}{\left(TP+FN\right)}$$

### 3.6. Confusion Matrix

A confusion matrix is a table that compares the number of predicted classes in a training against the number of correct classes (called ground truth classes). The confusion matrix so reflects how well the network performs for each class.
If the model successfully predicted the class for a sample, it is a true positive (TP).If the model successfully predicted another class for a sample that did not belong to the class, the result is a true negative (TN).If the model mistakenly predicted the class for a sample that actually belongs to another class, it is called a false positive (FP).If the model mistakenly predicted another class for a sample that truly belongs to that class, the result is a false negative (FN).The ground truth, or the number of correspondingly labeled images, is contained in the confusion matrix’s columns. The expected results for each class are organized in rows. The confusion matrix is shown in Table 6 and is graphically shown in Figure 21.Correct Predictions Evaluation: The accurately predicted images per class are represented by the numbers on the main diagonal (shown white on green) shown in Figure 21. As a result, all occurrences that are not on the major diagonal signal that the forecast is incorrect.Identifying False Positive Predictions: The FP column represents the number of images that have been predicted to belong to this class but actually belong to another class for each row, i.e., each class to be predicted.Evaluating False Negative Predictions: The FN row displays the number of images for which the network failed to detect that the images belong to this class for each column, i.e., each ground truth label.

#### Confidence Intervals for Accuracy, Precision, Recall, and F1 Score

Confidence intervals (CIs) provide a range of values within which the true performance metric (i.e., accuracy, precision, recall, or F1 score) will lie with a specified probability (i.e., 95% confidence level). It is essential to calculate CIs for these metrics to give an estimate of uncertainty in model performance.

The bar chart illustrated in Figure 22 shows classification performance metrics like accuracy, precision, recall, and F1 score derived from the provided confusion matrix. Each bar represents the metric’s value, with error bars indicating 95% confidence intervals, offering insight into the reliability of these estimates. The model exhibits exceptionally high performance, with all four metrics approaching 1.0, indicating strong classification accuracy, minimal false positives, and low false negatives. The narrow confidence intervals suggest high statistical certainty and consistency in performance. Notably, the precision and recall are nearly identical, reflecting the model’s balanced ability to correctly classify positive cases while minimizing false positives.

### 3.7. Loss Function

The loss function has the goal of optimization; its value over time reflects the training’s overall progress. The stated value of the loss function should decrease as the training progresses, as shown in Figure 23. This implies that the difference between the network’s anticipated outcome and the ground truth should decrease. Otherwise, the training will diverge, and it will have to be repeated with different settings, such as a slower starting learning rate or a larger batch.

### 3.8. Top-1 Error Graph

The “Top-1 Error” graph depicts the number of images where the forecasting class is incorrect. The stated value of the Top-1 error should decrease over time as the training advances. The training may need to be resumed with modified parameters if this is not the case. The Top-1 error graph of the EANN is shown in Figure 24. To avoid over- or underfitting, keep an eye on the progress of the training error in relation to the validation error. If the model continues to minimize the error on the training set, but the error on the validation set increases due to overfitting, make sure the amount of accessible data and their quality are appropriate. Reduce the learning rate hyperparameter if the error on the training set does not improve significantly due to underfitting.

## 4. Solar Module Image Classification Using the Compact Neural Network

A CNN typically refers to a compact neural network architecture that has been optimized to be smaller and more efficient while maintaining a high level of performance [37]. Compact CNNs are designed to reduce computational complexity, memory usage, and inference time, making them suitable for deployment on devices with limited resources, such as embedded systems, mobile devices, and edge computing platforms [34,38]. By using techniques like depthwise separable convolutions, pruning, and quantization, they achieve a balance between performance and resource usage. The fundamental tendency for achieving improved accuracy in major tasks such as image classification is to create deeper and wider CNNs. Compact networks can achieve excellent accuracy while using a modest quantity of processing resources. To boost computational efficiency, these networks are more likely to use “sparsely connected” convolutions like group convolutions rather than typical “completely connected” convolutions.

The detailed structure of the compact NN operation is shown in Figure 25. We divide a layer’s input feature maps into multiple groups; the first group’s output feature maps are concatenated with the second group’s input feature maps and then fed to the filters. This method is repeated until all of the input feature maps have been incorporated. However, the MVTec HALCON deep learning tool provides a pre-trained compact NN.

The size of convolutional filters in modern deep neural networks is usually 3 × 3 or 1 × 1, and the convolutional layer is the main computational expense, so the fully connected layer can be considered a special instance of the 1 × 1 convolutional layer. A very efficient approach for reducing convolution parameters is to replace ordinary 3 × 3 convolution with a 3 × 3 depthwise separable convolution. This approach reduces the model size dramatically, attracting more attention as a result.

Because the 1 × 1 filters cannot be separated, group convolution emerges as a promising and practical option that works well with a variety of lightweight deep neural network architectures.(6)$$Y_{i}=\left\{\begin{array}{c} X_{i}\times W_{i\;}i=1\\ concatenate\;\left(X_{i},Y_{i-1}\right)\times Wi\;1<i\leq G \end{array}\right.$$

In general, a typical 1 × 1 convolutional layer uses the filters W ∈ R O × I ×1 × 1 to turn the input feature maps X ∈ R I × Hin × Win into the output feature maps Y ∈ RO × Hout × Wout. The numbers I and O refer to the number of input and output feature maps, respectively. G groups are used to split the input channels and filters, where * denotes the convolutional operation 1 × 1

### 4.1. Dataset

The solar module datasets are composed of RGB images with a size of 224 × 224 pixels that belong to four different classifications. The label statistics are shown in Figure 26. There are 2534 training photos and 519 test images in the datasets. The images are zero-padded with 4 pixels on each side, randomly cropped to yield 32 × 32 images, and horizontally mirrored with a probability of 0.5 using a typical data augmentation approach.

### 4.2. Training Evaluation

The training evaluation of the compact NN is shown in Table 7 and Table 8. The accuracy of the model is defined as the number of correct predicted images over the total images, so the accuracy of the model is 99.91%. The top 1 error is a top-k error variant. It specifies the number of images for which the predicted classification is incorrect. The top error of the model is 0.09%. The harmonic mean and precision are called the F1 score. Precision is a measure of how accurate the model is at detecting positive samples. Recall is a metric that evaluates how well a model can detect positive samples. The precision and recall of the compact neural network model are 99.95% and 99.94%, respectively.

### 4.3. Confusion Matrix of Compact NN Model

A confusion matrix thatcompares the number of predicted classes in a training against the number of correct classes (called ground truth classes). It demonstrates how well the network performs for each class. The ground truth, or the number of correspondingly labeled images, is contained in the confusion matrix’s columns. The expected results for each class are stored in rows. The confusion matrix of the compact NN is shown in Figure 27.

#### Confidence Intervals for CNN Performance Metrics

A key observation across all graphs was the narrow confidence intervals, signifying high statistical reliability and stability in performance. The precision and recall values were nearly equal, showing that the models maintained a balanced trade-off between identifying true positives and minimizing false positives, with the F1 score reinforcing this balance as shown in Figure 28. Moreover, the accuracy was consistently close to 1.0, indicating that very few predictions were incorrect. The small error bars further confirmed that the models’ performance remained highly consistent and robust across different datasets. Overall, these results indicate that the classification models are highly effective, stable, and reliable, making them well suited for precise classification tasks with minimal risk of misclassification.

### 4.4. Loss Function and Top 1 Error Graph of Compact NN

The loss function of a Top-1 error graph of the compact neural network is shown in Figure 29 and Figure 30. During the training phase, a function is optimized. It penalizes deviations by comparing the network’s prediction with the given information about what it should detect in the image. When there is a Top-1 error, we look to see if the target label matches the most likely forecast.

### 4.5. Experimental Result Comparison

To compare the results of the methods listed in Table 9, the achieved model accuracy and other important indicators of model accuracy were used. However, these pre-trained models are available in MVTec HALCON. The compact ANN is more accurate as well as low-cost due to the single-layer model; moreover, every model has its own pros and cons. However, if the available data are abundant, then the need to implement an enhanced artificial neural network is high for dense architectures.

## 5. Mechanical Design for Solar Module Cleaning System

The mechanical design for a solar module cleaning system using HALCON integrates a robotic-based cleaning mechanism with HALCON’s powerful image processing capabilities to create a smart, autonomous solution. By detecting soiled regions, adjusting cleaning patterns, and optimizing energy and water use, the system ensures that solar panels are cleaned efficiently, maximizing energy output. Figure 31 shows a complete 3D design of the solar module cleaning system with the labels. Each label part is enlisted in Table 10.

### Electrical Design

This section provides the electrical design of the solar module cleaning system (SMCS) that we used for verifying the operations of the designed SMCS. In Figure 32, the operational flow and system component integration is shown.

Solar panel cleaning involves various processes that are very complex, but this simple and easy cleaning system is completely based on a deep learning and computer vision robotic system. The development of such a machine-based solar panel cleaning system is very cheap as well, and is controlled by sensors and motors to clean the panel efficiently. Figure 30 depicts the subsystems that will be used in our project. The mechanical subsystem and the control subsystem are illustrated in the block diagram as the two main subsystems. The cleaner as well as the DC motors and the stand that supports the solar panel, rechargeable battery, and water pump are all part of the mechanical subsystem. The relay, solar charger, Jetson nano board, and microcontroller are all found in the control subsystem [39,40,41,42].

These robots do not have centralized decision-making or onboard logic, and they are programmed to go to a specific place, such as a clean environment. When utilized in conjunction with integrated algorithms, such as computer vision technologies or convolutional neural networks, as well as the Internet of Things (IoT), the robotic platform can be extremely effective [36,37]. However, there are still some challenges, like time scheduling for the cleaning system and sensor reliability. A comparison of different solar module cleaning robots is shown in Table 11. The operational workflow is carried out according to the following steps:Solar cells are layers of semiconductive materials that create an electrical current when exposed to light. In this system, a solar module is installed on the top base of the robot, which provides power to the battery through a charge controller.The solar panel and battery are controlled by the solar charge controller, which also serves as the data source. It has a flexible charge algorithm, as well as over-temperature protection and power de-rating when the temperature rises. It displays the voltage and efficiency of the solar panel in the time domain to the user.First, the VE must be connected. A Bluetooth dongle connected to a Raspberry Pi reads data from the charger. In actuality, the data generated by the MPPT controller are Transistor–Transistor Logic (TTL) communication, which the vendor provides and explains. It is a service that gives weather data to online service and mobile application developers, including current weather data, forecasts, and historical data.A web camera is adopted for detecting and recognizing the soiling on the panel and sending the images to a Jetson nano board (CPU + GPU).The robot’s controller, which includes a CPU (central processing unit), GPU (graphics processing unit), and motor driver, is like a human brain. The CPU handles all the algorithms, including finding, image preprocessing, and path planning. The GPU is used to run the soiling recognition and segmentation algorithm. For image processing and deep learning computations, the GPU has a parallel structure that makes it more efficient than a standard CPU.The servomotor, which is controlled by a motor driver when the control system sends a signal to the drive motor, starts and moves along the panel and cleans it.When less water is required, the motor pump starts, and water fills in a tank that is erected on the robot.The pre-trained model and MVTec HALCON library installed on the Jetson nano board are used to analyze the soiled images acquired from the camera.

## 6. Discussion and Analysis

The results of this study highlight the effectiveness of deep learning-based classification for detecting soiling on PV modules. EANN and compact CNN models were evaluated for their accuracy, computational efficiency, and real-time applicability in automated PV maintenance systems. The proposed approach successfully integrates deep learning with machine vision software, improving classification accuracy while reducing unnecessary cleaning cycles. This section provides a comparative performance analysis, a discussion of practical implications, and an evaluation of potential challenges in implementing intelligent solar panel maintenance solutions.

### 6.1. Performance Comparison of EANN and CNN Models

The classification accuracy achieved by both EANN and compact CNN models was exceptionally high, demonstrating their capability in distinguishing between clean and soiled PV modules with minimal misclassification. The EANN model achieved an overall accuracy of 99.76%, with an F1 score of 99.85%, indicating strong model generalization. Additionally, precision and recall values of 99.87% and 99.83%, respectively, confirm the model’s robustness in correctly identifying soiled and clean panels. However, the inference time per image for the EANN was 15.73 ms, which makes it computationally demanding for real-time applications, particularly for embedded systems used in autonomous cleaning robots.

In contrast, the compact CNN model demonstrated superior computational efficiency, achieving an accuracy of 99.91%, an F1 score of 99.95%, and precision and recall scores of 99.95% and 99.94%, respectively. The compact CNN had a significantly lower inference time of 4.99 ms per image, making it highly suitable for real-time processing in autonomous cleaning robots that require quick decision-making. The superior performance of the CNN over EANN can be attributed to several key factors:Architectural Differences: CNNs leverage convolutional layers to efficiently extract hierarchical spatial features from images. Unlike EANNs, which rely on fully connected layers processing raw pixel data, CNNs apply local feature detectors (kernels) that enhance spatial pattern recognition. This enables CNNs to learn meaningful soiling patterns more effectively, leading to improved classification accuracy.Training Data Utilization: Both models were trained on the same dataset, but CNNs naturally perform better when handling spatially correlated image data. The ability to retain local features through pooling layers ensures CNNs can focus on important visual characteristics while reducing unnecessary computational overhead, unlike EANNs, which process all pixel values without prioritizing spatial dependencies.Hyperparameter Optimization: The CNN model likely benefited from hyperparameter tuning strategies, such as optimized learning rates, weight regularization, and dropout layers. Dropout layers help prevent overfitting, while batch normalization enhances training stability. In contrast, EANNs require careful tuning of weight initialization and activation functions to prevent vanishing gradient issues, which may have contributed to their slightly lower efficiency.

These factors collectively explain why the compact CNN model outperformed the EANN model in some respects, particularly in terms of inference time and computational efficiency. Given these findings, CNN-based architectures are more suitable for real-time deployment in solar panel maintenance systems due to their lower computational costs and faster processing speeds.

### 6.2. Confusion Matrix Analysis

The confusion matrix results further validate the reliability of both models. The EANN model misclassified only 1 out of 419 test images, demonstrating high classification reliability, particularly in distinguishing between clean and soiled PV panels. Similarly, the compact CNN model had only three false positives in a dataset of 3470 images, reinforcing its high accuracy and minimal error rate. The misclassifications observed were primarily in distinguishing between partially malfunctioned panels and clean panels, which could be attributed to subtle differences in soiling intensity and variations in lighting conditions during image acquisition.

The analysis suggests that both models are highly effective in accurately classifying soiling levels, but the compact CNN’s lower inference time makes it a more practical choice for real-time applications. This trade-off between classification accuracy and computational efficiency is essential when selecting a model for deployment in real-world PV maintenance systems, where real-time performance and power consumption constraints must be considered.

### 6.3. Comparison with Existing Studies

Several prior studies have explored machine learning- and deep learning-based soiling detection, but they typically rely on traditional CNN architectures or conventional image processing techniques without optimization for real-time deployment. Many existing methods still face challenges in computational efficiency, scalability, and cost-effectiveness, which this study aims to address. Compared to previous approaches, the proposed system introduces the following improvements:Integration of HALCON Machine Vision Software: Unlike many prior studies that focus solely on CNN-based classification, this study integrates HALCON software for advanced image preprocessing, segmentation, and feature extraction. This enhancement improves detection accuracy and efficiency, making it more suitable for real-world PV maintenance applications. Previous research often relied on basic image processing techniques, which are less adaptable to varying environmental conditions.Improved Real-Time Performance and Computational Efficiency: Many previous studies require high computational resources, making their models less suitable for real-time deployment in edge devices or autonomous PV cleaning robots. The compact CNN model proposed in this study achieves a significantly lower inference time (4.99 ms per image) while maintaining high classification accuracy (99.91%), making it more practical for real-time applications. In contrast, conventional CNN-based models often suffer from slow processing times, limiting their feasibility for large-scale deployment.Comparison with Traditional and Advanced Methods: Traditional threshold-based image processing techniques, widely used in earlier research, struggle under varying lighting conditions and different soiling patterns, leading to lower generalization capability. The proposed deep learning-based approach ensures higher adaptability and robustness across different environmental conditions. Compared to other advanced AI-driven soiling detection methods, such as sensor-based detection systems, our approach eliminates the need for additional hardware, such as infrared cameras or dust particle sensors, making it a cost-effective solution. Previous research has explored AI-based sensor fusion approaches, but these methods often require expensive additional equipment, which increases deployment costs. By relying solely on RGB image-based classification, the proposed system reduces both hardware costs and maintenance expenses while maintaining high detection accuracy.Scalability and Practical Deployment: Previous studies focusing on deep learning for PV soiling detection have often been conducted in controlled laboratory environments, with limited evaluation on real-world solar farms. In contrast, our system is designed with scalability in mind, ensuring that it can be implemented in autonomous cleaning robots without requiring excessive computational resources. The ability to deploy the model on low-power edge devices enhances its feasibility for large-scale solar farms, addressing a gap in existing research.

By integrating real-time soiling detection, deep learning-based classification, and machine vision software, this study bridges the gap between research and practical implementation, providing a cost-effective, scalable, and efficient solution for solar panel maintenance.

### 6.4. Practical Implications for PV Maintenance

The integration of deep learning-based soiling detection with autonomous PV cleaning robots offers significant economic, environmental, and operational benefits. Traditional solar panel cleaning systems operate on fixed schedules, resulting in unnecessary water and energy consumption. The proposed method allows for real-time, targeted cleaning based on actual soiling levels, ensuring that only heavily soiled panels are cleaned while avoiding unnecessary maintenance. One of the most critical advantages of this system is its potential to improve energy efficiency. Studies have shown that soiling can lead to energy losses of up to 50% in heavily dust-affected regions. By implementing a deep learning-based classification system, solar farms can maintain peak energy output while extending the lifespan of PV panels. The ability to analyze PV module images in real time ensures that cleaning operations are conducted only when necessary, optimizing both operational efficiency and sustainability.

Furthermore, the low inference time of the compact CNN model enables seamless integration with large-scale solar farms, where thousands of panels need to be monitored continuously. The ability to process and classify images quickly ensures that cleaning actions can be automated efficiently, reducing reliance on manual inspection and maintenance teams. This is particularly beneficial for remote solar farms where human intervention is costly and time-consuming. The results of this study highlight several important findings. The EANN and compact CNN models achieved high classification accuracy, with the compact CNN offering superior inference speed (4.99 ms per image), making it ideal for real-time deployment. The integration of HALCON software improved image preprocessing and classification, enhancing the system’s efficiency and scalability.

The proposed approach enables targeted cleaning of soiled PV modules, optimizing water and energy consumption for sustainable solar farm management. This study also identified key deployment challenges, including computational constraints in edge devices and environmental variability, which must be addressed to improve system robustness. By overcoming these challenges, deep learning-based PV maintenance systems can be effectively deployed for large-scale solar energy production.

## 7. Conclusions

This study proposed a deep learning-based classification system to enhance the maintenance of PV modules by automating the detection and classification of soiling levels. The system integrated compact CNNs and an EANN with HALCON machine vision software to enable real-time identification of soiled PV panels. By leveraging deep learning and computer vision, the system enhances the efficiency of autonomous solar cleaning robots, allowing for them to selectively clean heavily soiled areas instead of following fixed schedules. This targeted cleaning approach optimizes resource utilization, reduces unnecessary water and energy consumption, and maintains or improves solar power efficiency. The experimental results confirmed the high accuracy and computational efficiency of the proposed models, demonstrating their suitability for real-time deployment in large-scale solar farms. The findings showed the potential of AI-driven PV maintenance solutions to improve operational efficiency, reduce maintenance costs, and extend the lifespan of solar installations. The conclusions of this study can be summarized in the following points:The proposed deep learning-based classification system successfully integrates CNNs and an EANN with HALCON software, enabling real-time and accurate soiling detection for PV modules. The system automates image processing and classification, reducing dependency on manual inspections.The compact CNN model achieved an accuracy of 99.91%, while the EANN reached 99.76%, both demonstrating high precision and recall scores. The compact CNN had a lower inference time of 4.99 ms per image, compared to 15.73 ms for the EANN, making it more efficient for real-time processing in autonomous cleaning robots.The system enables targeted cleaning, ensuring that only heavily soiled panels are cleaned, which reduces unnecessary maintenance cycles and optimizes water and energy consumption in solar farms. This approach helps maintain high energy output and system efficiency over time.The automation of PV maintenance through AI-driven cleaning robots reduces the need for manual intervention, lowers maintenance costs, and improves scalability. The proposed method proves feasible for large-scale solar farms, where thousands of PV panels require continuous monitoring and cleaning.The findings demonstrated that AI-driven PV maintenance systems enhance operational efficiency, reduce costs, and extend the lifespan of solar installations. This research provides a foundation for future advancements in AI-based solar maintenance, highlighting the practical application of deep learning and machine vision in optimizing solar energy management.

## Figures and Tables

**Figure 1 sensors-25-01295-f001:**
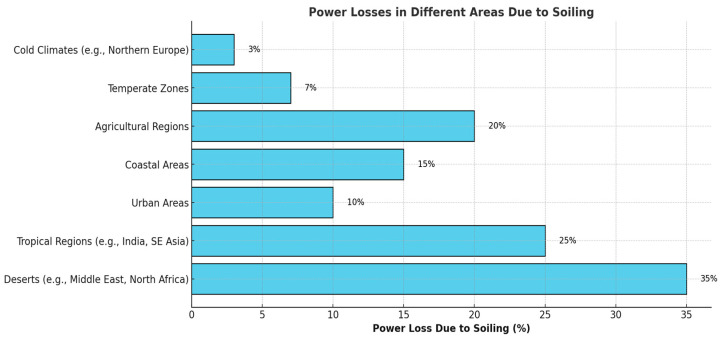
Solar power losses due to soiling in different areas.

**Figure 2 sensors-25-01295-f002:**
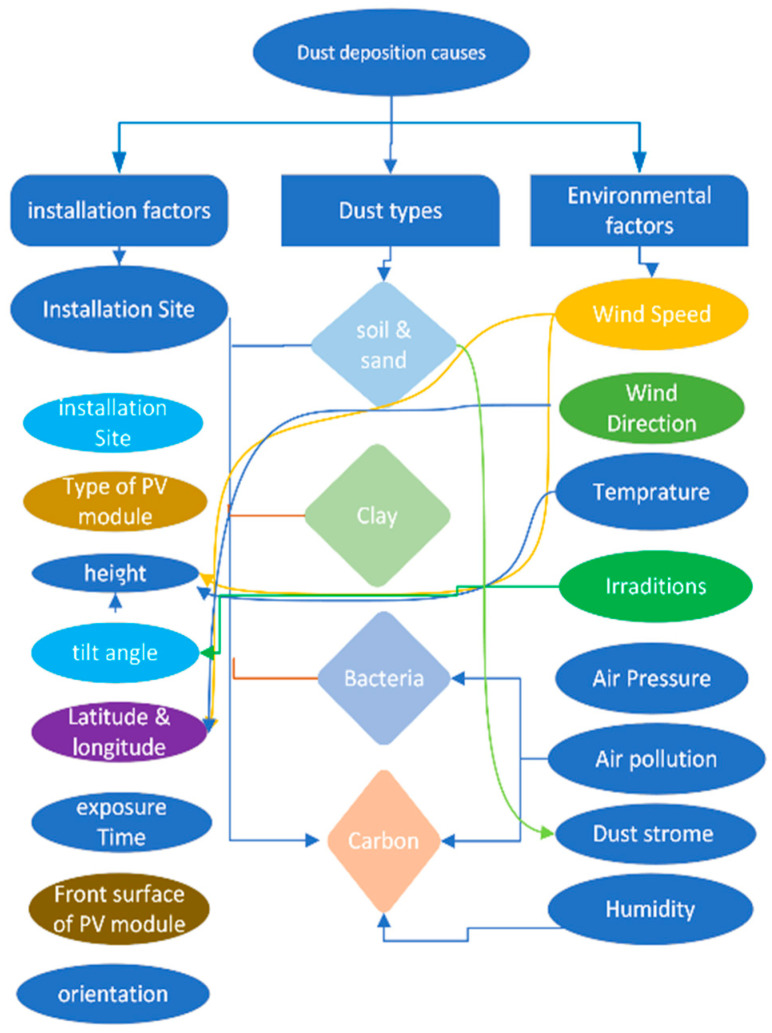
Interlink between various sources of dust accumulation on solar panels.

**Figure 3 sensors-25-01295-f003:**
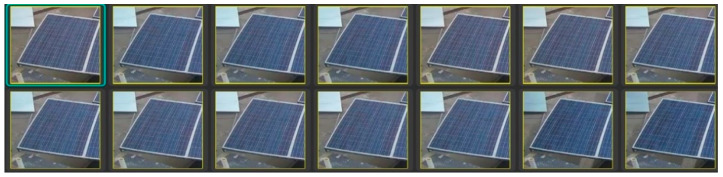
Clean solar panel labeled dataset.

**Figure 4 sensors-25-01295-f004:**
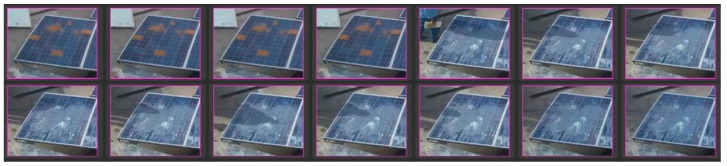
Partially malfunctioned solar panel labeled dataset.

**Figure 5 sensors-25-01295-f005:**
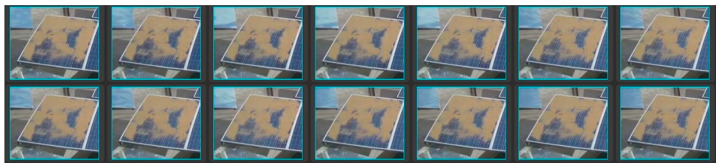
Highly malfunctioned solar panel labelled dataset.

**Figure 6 sensors-25-01295-f006:**
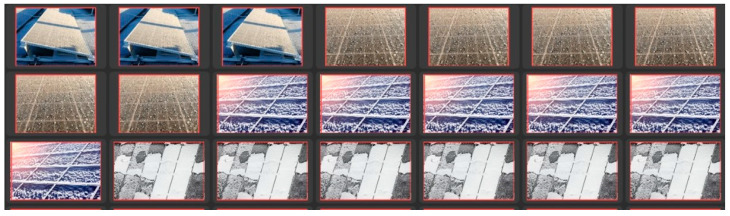
Snow-malfunctioned solar panel labelled dataset.

**Figure 7 sensors-25-01295-f007:**
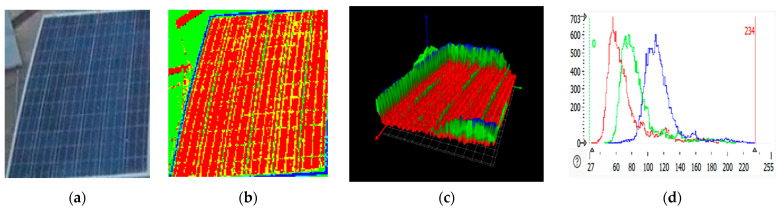
Clean solar panel image processing. (**a**) solar panel. (**b**) RGB representation. (**c**) 3D plots. (**d**) RGB histogram.

**Figure 8 sensors-25-01295-f008:**
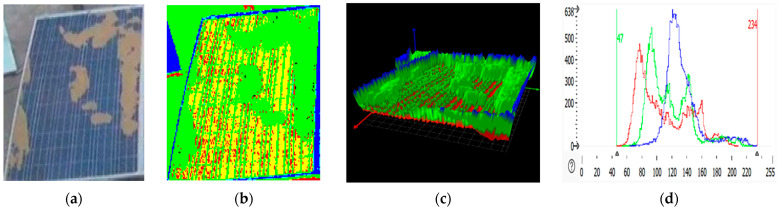
Partially malfunctioned solar panel image processing: (**a**) solar panel; (**b**) RGB representation; (**c**) 3D plot; (**d**) RGB histogram.

**Figure 9 sensors-25-01295-f009:**
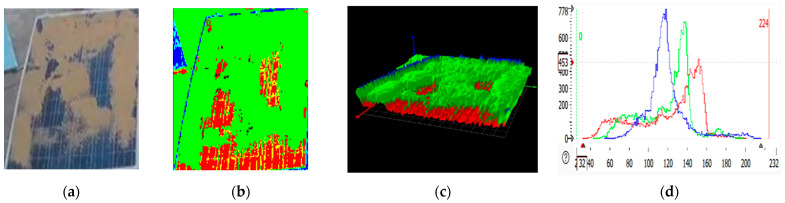
Highly malfunctioned solar panel image processing: (**a**) solar panel; (**b**) RGB representation; (**c**) 3D plot; (**d**) RGB histogram.

**Figure 10 sensors-25-01295-f010:**
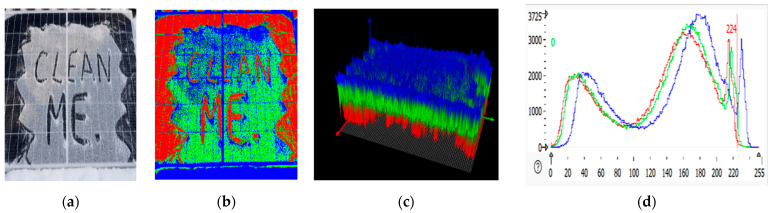
Snow-malfunctioned solar panel image processing: (**a**) solar panel; (**b**) RGB representation; (**c**) 3D plot; (**d**) RGB histogram.

**Figure 11 sensors-25-01295-f011:**
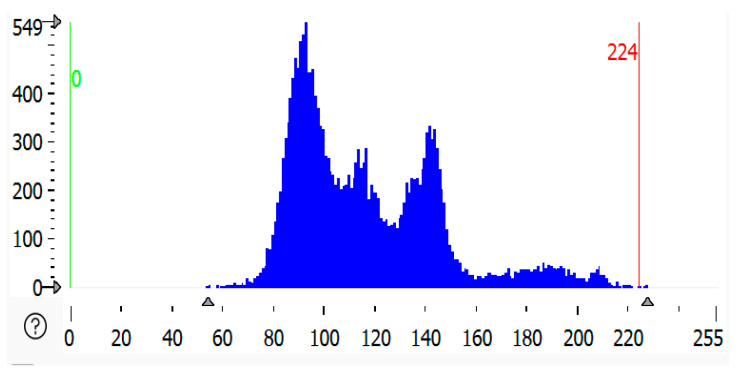
Histogram of partially malfunctioned region of solar panel.

**Figure 12 sensors-25-01295-f012:**
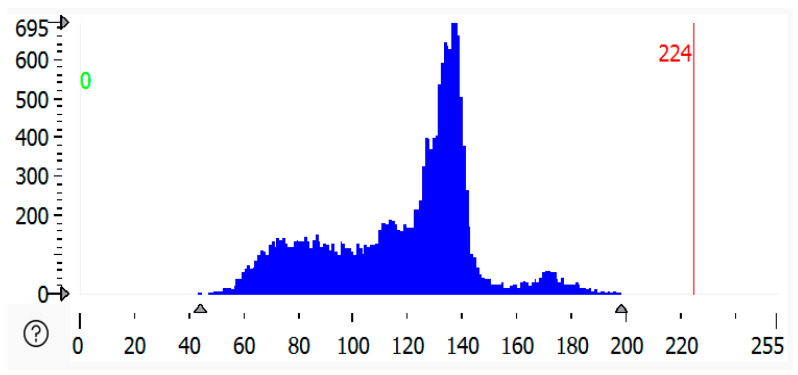
Histogram of highly malfunctioned solar panel.

**Figure 13 sensors-25-01295-f013:**
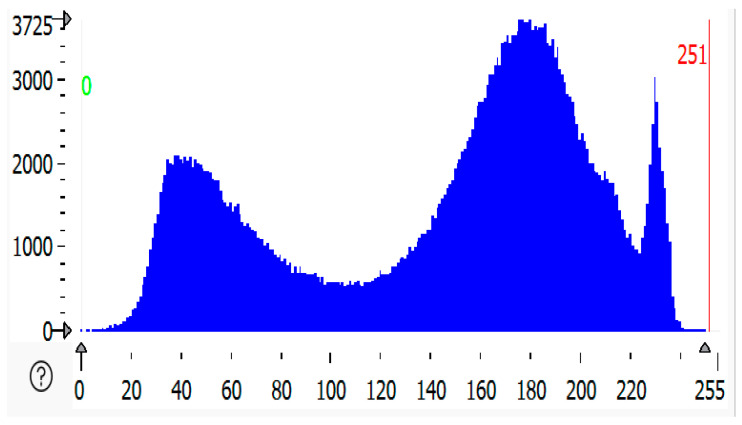
Histogram of snow-malfunctioned solar panel.

**Figure 14 sensors-25-01295-f014:**
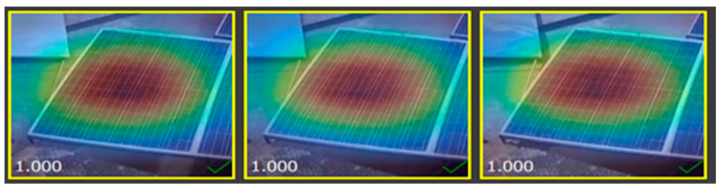
Heat map of clean solar panel.

**Figure 15 sensors-25-01295-f015:**
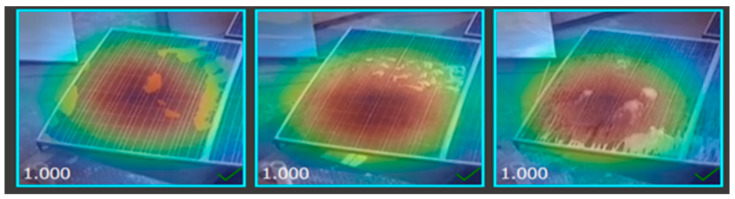
Heat map of partially malfunctioned solar.

**Figure 16 sensors-25-01295-f016:**
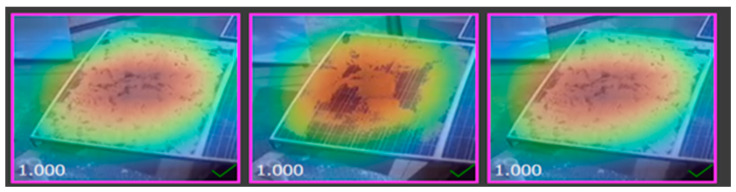
Heat map of highly malfunctioned solar panel.

**Figure 17 sensors-25-01295-f017:**
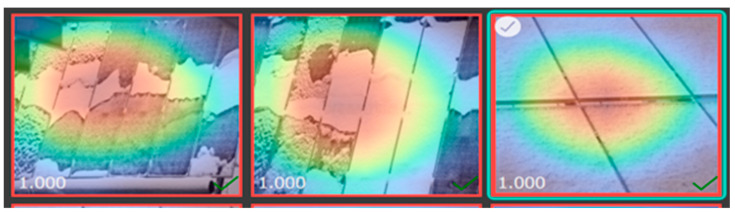
Heat map of snow-malfunctioned solar panel.

**Figure 18 sensors-25-01295-f018:**
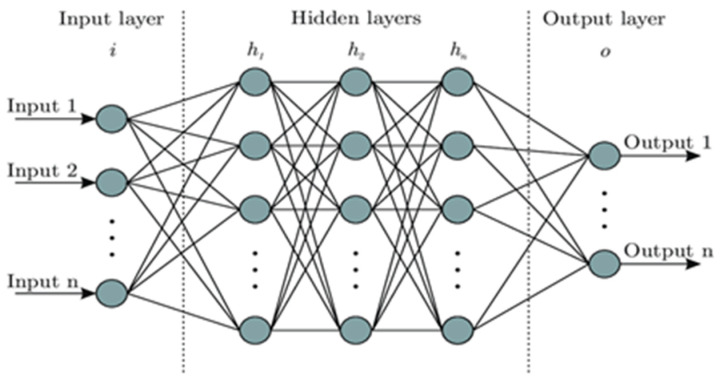
Enhanced artificial neural network structure.

**Figure 19 sensors-25-01295-f019:**
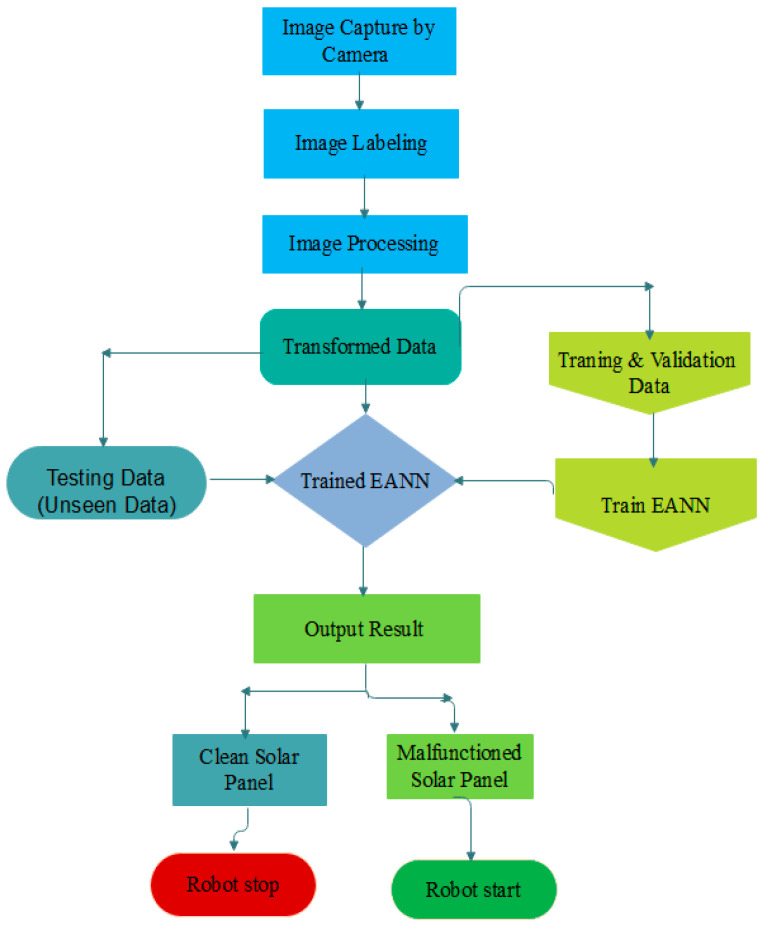
Working flowchart of EANN.

**Figure 20 sensors-25-01295-f020:**
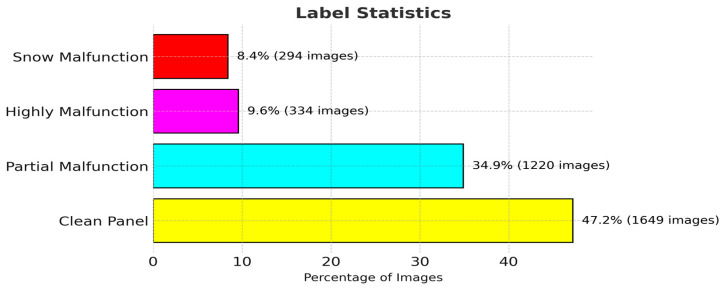
Label statistics of dataset.

**Figure 21 sensors-25-01295-f021:**
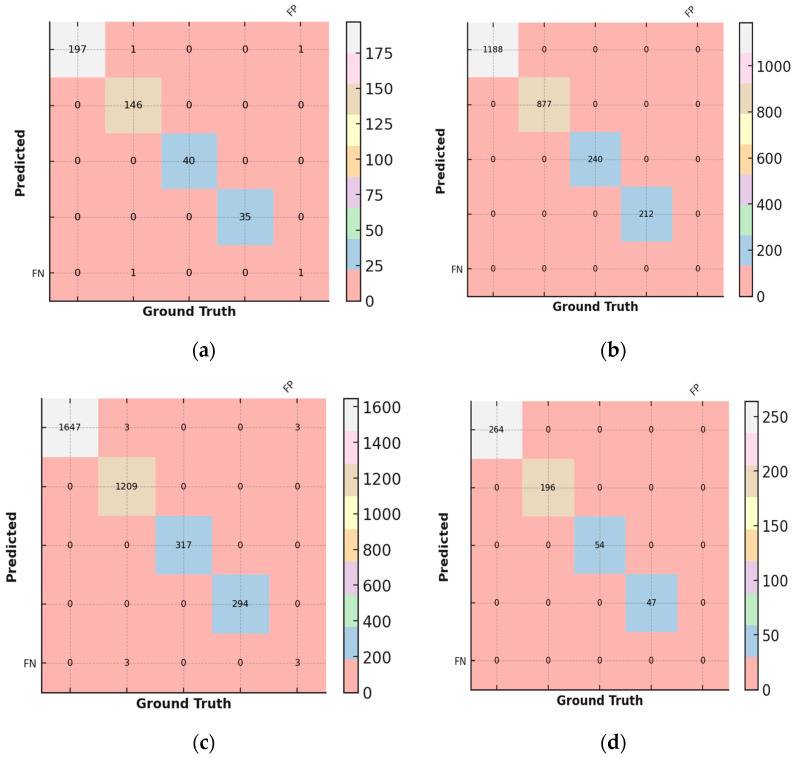
Representation of confusion matrix by different diagrams.

**Figure 22 sensors-25-01295-f022:**
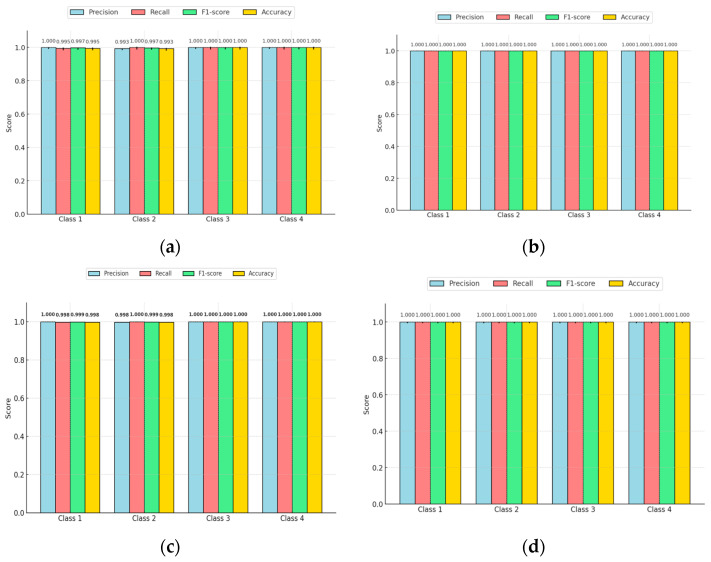
Representation of confidence intervals for precision, recall, F1 score, and accuracy of EANN confusion matrix.

**Figure 23 sensors-25-01295-f023:**
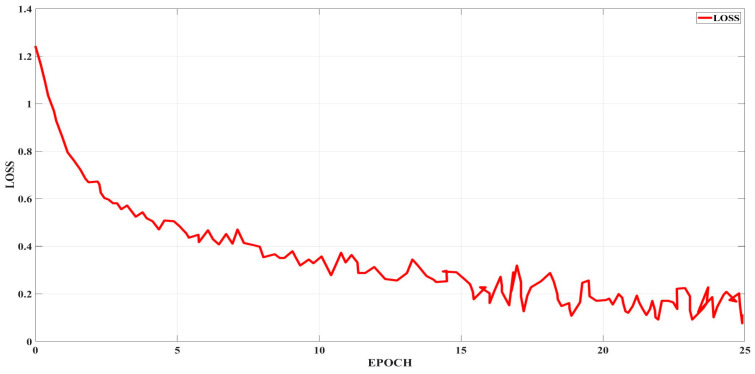
Loss function.

**Figure 24 sensors-25-01295-f024:**
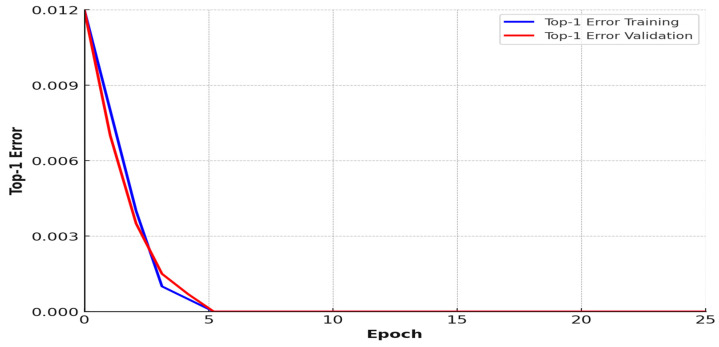
Top-1 error graph.

**Figure 25 sensors-25-01295-f025:**
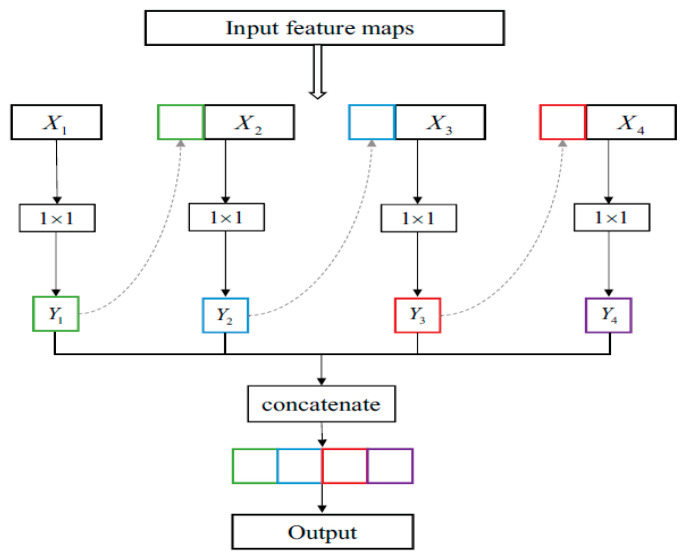
Operation of compact neural network.

**Figure 26 sensors-25-01295-f026:**
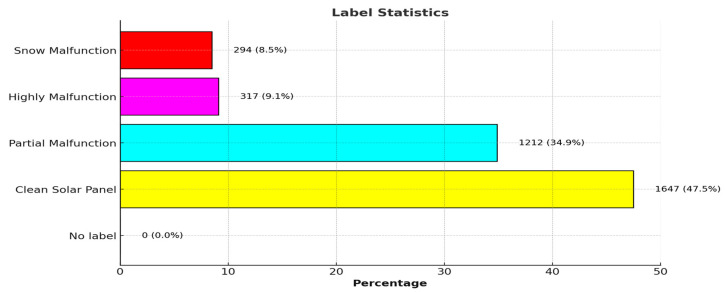
Statistical representation of compact NN model dataset.

**Figure 27 sensors-25-01295-f027:**
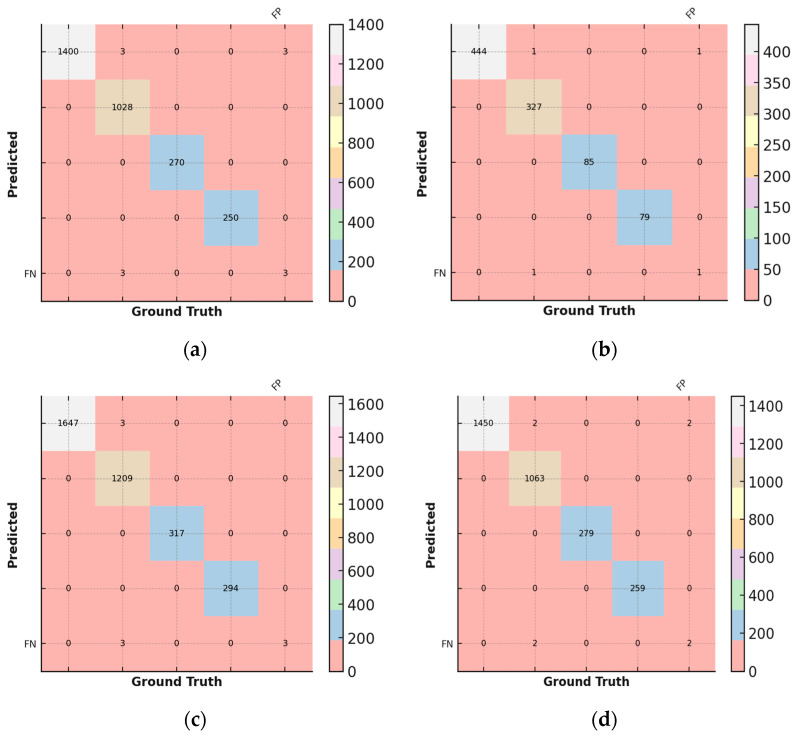
Different confusion matrixes of compact NN model.

**Figure 28 sensors-25-01295-f028:**
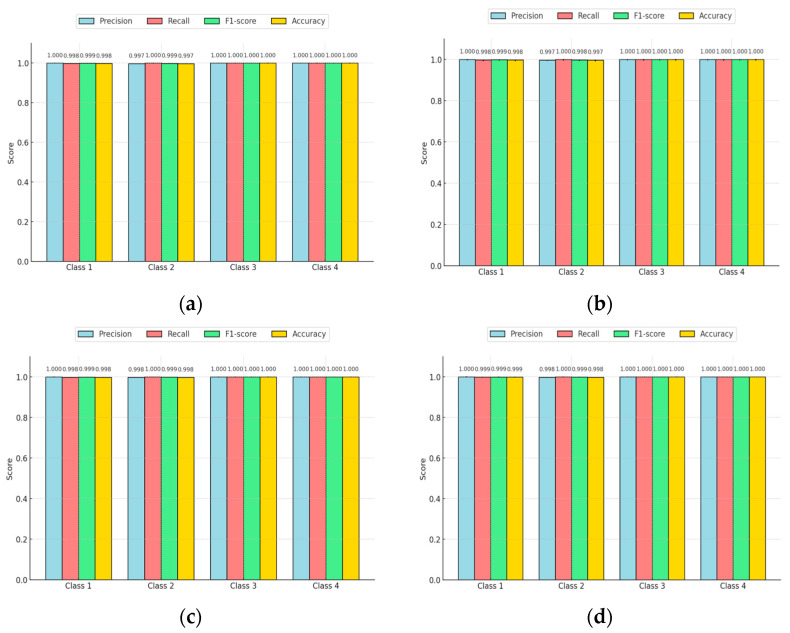
Confidence intervals for accuracy, precision, recall, and F1 score of confusion matrix of compact NN model.

**Figure 29 sensors-25-01295-f029:**
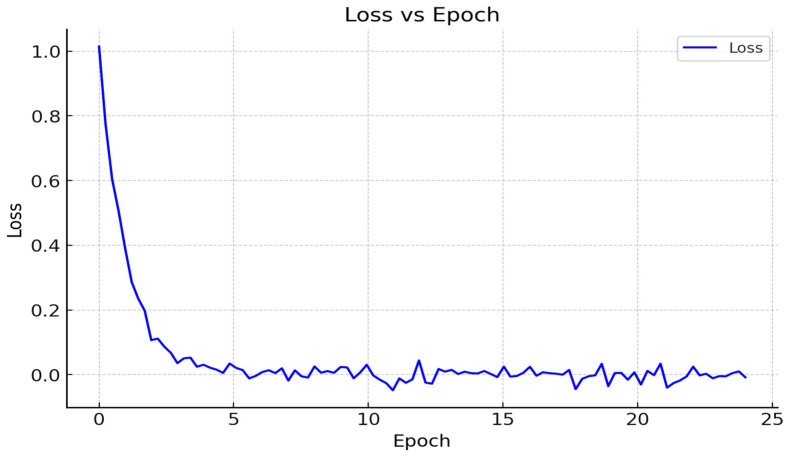
Loss function of compact NN.

**Figure 30 sensors-25-01295-f030:**
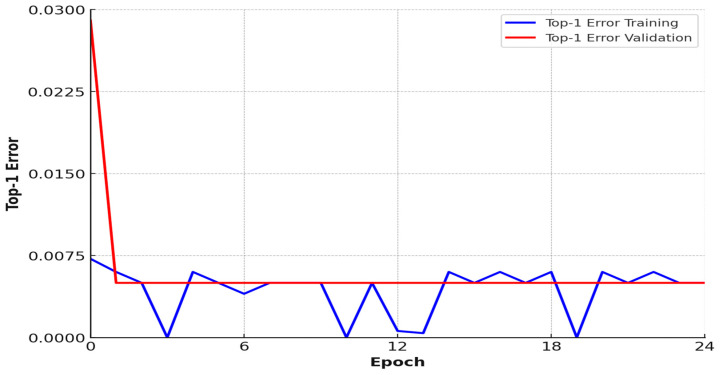
Top-1 error graph of compact NN.

**Figure 31 sensors-25-01295-f031:**
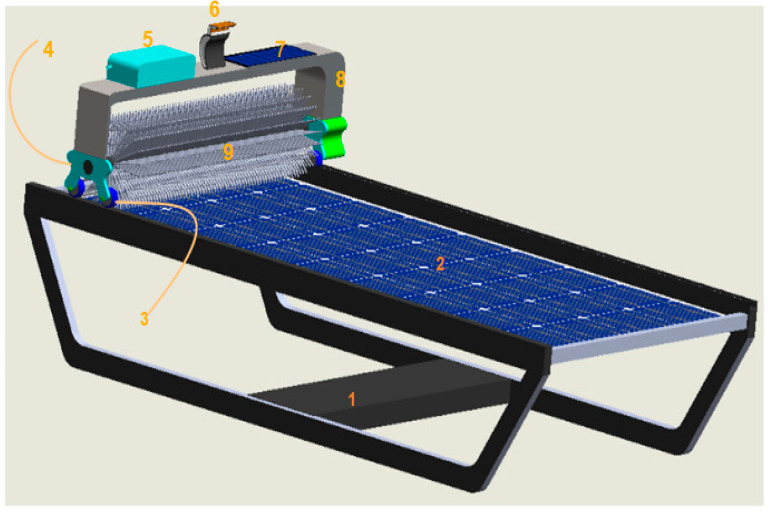
Three-dimensional view of solar panel cleaning system.

**Figure 32 sensors-25-01295-f032:**
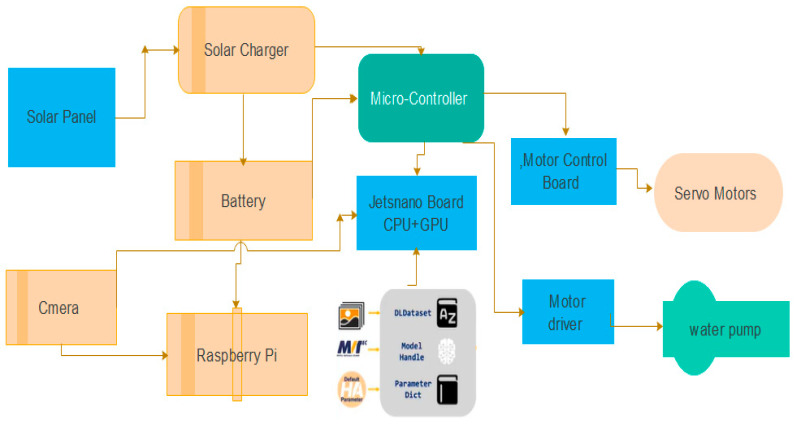
Integration of computer vision-based solar panel cleaning robot’s diagram.

**Table 1 sensors-25-01295-t001:** Statistical analysis of RGB histogram of solar panel.

Solar Panels	Grey Value Bytes (Peak, *X*-Axis)	Grey Value Frequency (Peak, *Y*-Axis)	Grey Value Range	Percentage
Clean solar panel	110	602	30–234	2.1198%
Partially malfunctioned solar panel	121	638	47–234	3.1213%
Highly malfunctioned solar panel	118	778	33–216	4.1194%
Snow-malfunctioned solar panel	117	3725	0–251	3.0347%

**Table 2 sensors-25-01295-t002:** Statistical analysis of grey histogram of malfunctioned region of solar panel.

Solar Panels	Gray Value Bytes (Peak, *X*-Axis)	Gray Value Frequency (Peak, *Y*-Axis)	Gray Value Range	Percentage
Partially malfunctioned solar panel	93	594	54–228	2.6859%
Highly malfunctioned solar panel	137	695	44–198	3.6799%
Snow-malfunctioned solar panel	177	3725	0–250	2.03472%

**Table 3 sensors-25-01295-t003:** Split information of dataset.

Split Type	Percentage	Number of Images
Train	71.98%	2517
Validation	16.04%	561
Test	11.98%	419

**Table 4 sensors-25-01295-t004:** Evaluation measures of model.

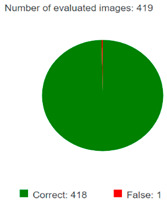 Number of evaluated images: 419 Correct: 418 False: 1	**Result Name**	**Value**
	Accuracy	99.76%
	Inference Time *	15.73 ms
	Preprocess Time *	0.46 ms
	Total Time *	16.19 ms
	Top-1 Error	0.24%
	F1 Score	99.85%
	Precision	99.87%
	Recall	99.83%

* Time per image.

**Table 5 sensors-25-01295-t005:** Class overview of model.

Class	FP	Precision [%]	FN	Recall [%]	F1 Score [%]	Number of Images
Clean panel	1	99.49	0	100	99.75	197
Partially malfunctioned panel	0	100	1	99.32	99.66	147
Highly malfunctioned panel	0	100	0	100	100	40
Snow-malfunctioned panel	0	100	0	100	100	35

**Table 6 sensors-25-01295-t006:** Confusion matrix.

	Clean Panel	Partially Malfunctioned Panel	Highly Malfunctioned Panel	Snow-Malfunctioned Panel	FP
Clean panel	**197**	**1**	0	0	**1**
Partially malfunctioned panel	0	**146**	0	0	**0**
Highly malfunctioned panel	0	0	**40**	0	**0**
Snow-malfunctioned panel	0	0	0	**35**	**0**
FN	0	1	0	0	1

**Table 7 sensors-25-01295-t007:** Training evaluation of compact neural network model.

Number of evaluated images: 3470 Correct: 3467 False: 3 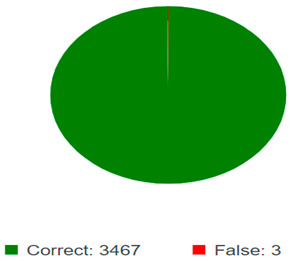	**Result Name**	**Value**
	Accuracy	99.91%
	Inference Time *	4.99 ms
	Preprocess Time *	0.51 ms
	Total Time *	5.5 ms
	Top-1 Error	0.09%
	F1 Score	99.95%
	Precision	99.95%
	Recall	99.94%

* Time per image.

**Table 8 sensors-25-01295-t008:** Class overview of compact neural network model.

Class	FP	Precision [%]	FN	Recall [%]	F1 Score [%]	Number of Images
Snow-malfunctioned solar panel	0	100	0	100	100	294
Clean solar panel	3	99.82	0	100	99.91	1647
Highly malfunctioned solar panel	0	100	0	100	100	317
Partially malfunctioned solar panel	0	100	3	99.75	99.88	1212

**Table 9 sensors-25-01295-t009:** Comparison of solar module image classification with DL methods.

Deep Learning Models	Training Dataset	Top-1 Error	F1 Score	Precision	Recall	Total Time	Accuracy
Enhanced artificial neural network (EANN)	Labeled dataset	0.24%	99.85%	99.87%	99.83%	16.19 m	99.76%
Compact neural network (CNN)	Labeled dataset	0.09%	99.95%	99.95%	99.94%	5.5 m	99.91%

**Table 10 sensors-25-01295-t010:** Solar module cleaning system with part labels.

SR NO	Name of Parts	Quantity Required
1	Solar stand	1
2	Solar panel	1
3	Roller and DC servo motor	4.4
4	Brush support part	2
5	Water storage tank	1
6	Camera	1
7	Solar plate and battery	1.1
8	Supporting frame for hole cleaning system	1
9	Cleaning brush	1

**Table 11 sensors-25-01295-t011:** Comparison of different robotic-based solar module cleaning systems.

Current Robotic based solar module cleaning Technologies	**Control Mechanism**	**Advantages**	**Disadvantages**
	A robot is controlled by an automated control system with a PLC	(1) It has the ability to carry out automatic cleaning, real-time network control, and equipment operating status monitoring. (2) It can be used with any type of panel.	(1) Sophisticated and complex system. (2) Restricted to particular module configuration.
	Sensors that detect dust accumulation and a DCR that accelerates on the PV panel	(1) Distant monitoring of real-time plant conditions. (2) Requires no human intervention. (3) Cleaning supervision can be carried out remotely.	(1) Complex system. (2) High initial investment. (3) Skilled and trained supervisor needed. (4) Time scheduling problem.
Computer Vision-based cleaning Robot (Proposed cleaning system)	This cleaning robot control system consists of a Jetson nano board (CPU + GPU) and a camera installed on the robot’s head that captures images of dirty solar panels, process and classifies the images using deep learning algorithms, and cleans the solar panel automatically.	(1) No labor cost. (2) High accuracy. (3) Real-time monitoring. (4) The process is much faster and more accurate than for other cleaning robots. (5) Increased overall PV system efficiency. (6) Low cost.	Large, labeled dataset required. Lack of standardization of solar panel. High computational cost.

## Data Availability

Data available in [30].

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
