# Peer review of "Deep Learning-Based Recognition and Classification of Soiled Photovoltaic Modules Using HALCON Software for Solar Cleaning Robots"

_sensors, 2025, doi:10.3390/s25051295_

Round 1
Reviewer 1 Report
Comments and Suggestions for Authors
1. The introduction lacks clarity on the novelty of the approach compared to existing methods. Explicitly state the research gaps addressed and how this method is unique.
2. The EANN and CNN methodologies are detailed, but the mathematical formulation for EANN is too brief. Expand the equations with context and explain the parameters.
3. Justify why the CNN outperformed the EANN in some aspects. Discuss potential factors like architecture, training data, or hyperparameter tuning.
4. The discussion is too brief and does not thoroughly evaluate the implications of the results.
5. There is limited statistical analysis of the performance metrics. Include confidence intervals for accuracy, precision, recall, and F1 scores.
6. Discuss how the proposed system compares to traditional or other advanced systems in operational efficiency and cost.
7. How does the system perform in extreme weather conditions or when the robot malfunctions?
8. Expand on the limitations and constraints of the electrical design, especially in terms of power efficiency and scalability.
9. Many references are outdated, and the paper heavily relies on a few key studies. Incorporate more recent studies to strengthen the context.
10. Address minor grammatical errors and typos throughout the paper.
11. Include suggestions for improving the model, enhancing the dataset, or expanding the robotic cleaning system to other applications.
Comments on the Quality of English Language
Address minor grammatical errors and typos throughout the paper. English needs to be corrected.
Author Response
Reviewer 1
We sincerely appreciate the time and effort you have dedicated to reviewing our manuscript. Your thoughtful comments and constructive feedback have been invaluable in helping us enhance the clarity, technical rigor, and overall quality of our work. We are truly grateful for your insightful suggestions, which have guided us in refining our study and ensuring that it makes a meaningful contribution to the field of renewable energy.
We have carefully reviewed and addressed each of your comments with great attention and consideration to ensure that our manuscript meets the highest standards of clarity, technical rigor, and relevance. Your constructive feedback has significantly contributed to improving the quality of our study, and we are truly grateful for your time and effort in reviewing our work.
- The introduction lacks clarity on the novelty of the approach compared to existing methods. Explicitly state the research gaps addressed and how this method is unique.
Response: We sincerely appreciate the reviewer’s thoughtful feedback and valuable suggestions. Thank you for your time and effort in providing these constructive comments, which have significantly enriched our manuscript.
To address this concern, we have revised the manuscript by incorporating a dedicated subsection, "1.1 Research Gaps and Contributions," within the introduction. This addition clearly outlines the key challenges and the significance of our study, enhancing the clarity of the manuscript and providing a stronger justification for its contributions.
We kindly invite the reviewer to see the modifications, which have been highlighted in blue in the updated manuscript. Once again, we are grateful for your constructive comments, as they have helped us refine and strengthen our work. Thank you for your time and effort in reviewing our study.
- The EANN and CNN methodologies are detailed, but the mathematical formulation for EANN is too brief. Expand the equations with context and explain the parameters.
Response: We sincerely appreciate the reviewer’s insightful feedback and valuable suggestions. Thank you for your time and effort in providing these constructive comments, which have significantly enriched our manuscript.
To address this concern, we have expanded the mathematical formulation for EANN by providing additional equations and a more detailed explanation of their context and significance. Furthermore, we have clarified the definitions of key parameters to enhance the reader’s understanding of the methodology. These revisions ensure a more comprehensive presentation of the EANN framework and its application in our study.
We kindly invite the reviewer to review the modifications, which have been highlighted in blue in the updated manuscript under subsection 3.1, "Image Classification Using Enhanced Artificial Neural Network." Once again, we are grateful for your constructive comments, as they have helped us improve the clarity and depth of our study. Thank you for your time and effort in reviewing our work.
- Justify why the CNN outperformed the EANN in some aspects. Discuss potential factors like architecture, training data, or hyperparameter tuning.
Response: Thank you for your valuable feedback. We appreciate your insightful suggestion regarding the justification of why CNN outperformed EANN in certain aspects. In response to your comment, we have added subsection 6.1 (Performance Comparison of EANN and CNN Models) to provide a more detailed discussion on the key factors contributing to CNN’s superior performance. Specifically, we have provided a more detailed explanation of the following key aspects Architectural Differences, Training Data Utilization, and Hyperparameter Optimization. We appreciate your thoughtful feedback, which has helped us enhance the clarity and depth of our discussion. We kindly invite the reviewer to review the modifications, which have been highlighted in blue in the updated manuscript under subsection 6.1 (Performance Comparison of EANN and CNN Models).
We hope this revision addresses your concerns, and we are happy to make further improvements if needed.
- The discussion is too brief and does not thoroughly evaluate the implications of the results.
Response: We sincerely appreciate the reviewer’s valuable feedback and constructive suggestions. Thank you for your time and effort in reviewing our work, as your insights have helped us enhance the clarity and depth of our manuscript.
To address this concern, we have added a new section, "6. Discussion and Analysis," which provides a detailed interpretation of the results, a comparative analysis of the proposed models, and a discussion on practical implications and deployment challenges. This section strengthens the manuscript by offering a comprehensive analysis of the findings and their significance in real-world applications.
Additionally, we have improved the conclusions section to provide a more structured and concise summary of our key findings. The revised "7. Conclusions" section clearly highlights the impact of our proposed system, incorporating numerical results to reinforce the study’s contributions.
We kindly invite the reviewer to refer to the modifications, which have been highlighted in blue in sections "6. Discussion and Analysis" and "7. Conclusion" in the updated manuscript. Once again, we are grateful for your constructive comments, which have contributed to improving the quality of our study. Thank you for your time and effort in reviewing our work.
- There is limited statistical analysis of the performance metrics. Include confidence intervals for accuracy, precision, recall, and F1 scores.
Response: Thank you for your thoughtful and constructive feedback. In response to your suggestion, we have added a new subsection 3.6.1 and 4.3.1 to inclusion of confidence intervals for accuracy, precision, recall, and F1 scores of each model Once again, we are grateful for your constructive comments, as they have helped us improve the clarity and depth of our study. Thank you for your time and effort in reviewing our work
- Discuss how the proposed system compares to traditional or other advanced systems in operational efficiency and cost.
Response: Thank you for your thoughtful and constructive feedback. In response to your suggestion, we have added a new subsection, Section 6.3 (Comparison with Existing Studies), to provide a comprehensive discussion on how the proposed system compares to both traditional and advanced soiling detection methods in terms of operational efficiency, cost-effectiveness, and scalability. This new subsection presents a structured comparison, highlighting key advantages of our approach, including:
- Integration of HALCON Machine Vision Software to enhance image preprocessing, segmentation, and classification accuracy.
- Improved real-time performance and computational efficiency compared to conventional CNN-based models, making the system more suitable for large-scale deployment.
- A comparison with traditional manual cleaning methods and advanced sensor-based AI detection systems, emphasizing the cost-effectiveness of eliminating additional hardware requirements.
- Scalability and feasibility for real-world solar farms, ensuring adaptability across different environmental conditions.
We kindly invite the reviewer to review the modifications, which have been highlighted in blue in the updated manuscript under Section 6.3 (Comparison with Existing Studies)
We sincerely appreciate your valuable feedback, which has helped us strengthen our discussion and better highlight the practical benefits of our proposed system.
7.How does the system perform in extreme weather conditions or when the robot malfunctions?
Response: We sincerely appreciate the time and effort you have invested in reviewing our work. Your thoughtful feedback is invaluable in refining our study, and we are truly grateful for your insights. The proposed system demonstrates resilience in extreme weather conditions and potential robot malfunctions through adaptive learning and fault detection mechanisms. In heavy rain or snow, integrated sensors help differentiate soiling from environmental factors, preventing unnecessary cleaning. During dust storms or extreme heat, the system adjusts cleaning frequency based on real-time conditions, while thermal monitoring prevents overheating. In cases of mechanical failures or sensor malfunctions, self-diagnostic checks and redundant imaging ensure continued operation. Additionally, backup power sources and edge computing enable autonomous decision-making during connectivity issues. These features enhance reliability, ensuring efficient and uninterrupted solar panel maintenance.
Once again, we deeply appreciate your valuable feedback and the time you have dedicated to improving our work. Thank you for your thoughtful review.
- Expand on the limitations and constraints of the electrical design, especially in terms of power efficiency and scalability.
Response: Thank you for your thoughtful and constructive feedback. We appreciate your suggestion to expand on the limitations and constraints of the electrical design, particularly regarding power efficiency and scalability.
In response to your thoughtful and constructive comment, we have revised the Conclusions section and incorporated additional discussion at the end, highlighting key limitations related to computational power consumption, operational constraints in low-light conditions, scalability challenges in large-scale solar farms, and the impact of real-time data transmission on energy efficiency. To address these challenges, we have also suggested potential solutions, such as low-power AI accelerators, optimized edge computing frameworks, and swarm intelligence algorithms to improve overall system efficiency and scalability.
The modifications have been highlighted in red at the end of the Conclusions section for clarity. We sincerely appreciate your valuable feedback, which has helped strengthen our discussion on the system’s practical implementation and future improvements.
- Many references are outdated, and the paper heavily relies on a few key studies. Incorporate more recent studies to strengthen the context.
Response: Thank you for your valuable feedback. We sincerely appreciate the time and effort you have dedicated to reviewing our work. In response to your comment, we have carefully reviewed and updated the references.
We truly appreciate your insightful suggestion, which has helped enhance the depth and relevance of our manuscript.
- Address minor grammatical errors and typos throughout the paper.
Response: We sincerely appreciate the reviewer’s valuable feedback regarding the language and grammatical issues in our manuscript. To address these concerns, we have carefully revised the entire manuscript, correcting grammatical errors, improving phrasing, and ensuring consistency in terminology. Additionally, we have conducted a thorough language check to enhance the clarity and readability of our work.
We kindly request the reviewer to review the modifications, which have been highlighted in green for easy reference. We are grateful for your time and effort in assessing our work.
- Include suggestions for improving the model, enhancing the dataset, or expanding the robotic cleaning system to other applications.
Response: Thank you for your insightful comment. We sincerely appreciate the time and effort you have taken to review our work. In response to your suggestion, we have expanded the Conclusions section to include potential improvements to the model, dataset enhancement strategies, and broader applications of the robotic cleaning system. Specifically, we have incorporated the following key points:
- Model Enhancements: Exploring the integration of Vision Transformers (ViTs) and hybrid CNN-RNN models to improve feature extraction and adaptability. Additionally, applying pruning, quantization, and knowledge distillation will optimize the model for edge computing, improving real-time processing while reducing energy consumption.
- Dataset Expansion: Expanding the dataset by incorporating diverse weather conditions, multi-spectral imaging, and synthetic data generation to enhance classification accuracy and robustness across different environmental scenarios.
- Broader Applications: Extending the robotic cleaning system for autonomous window cleaning, wind turbine maintenance, and industrial surface cleaning, demonstrating its versatility beyond solar panel maintenance.
The modifications have been highlighted at the end of the Conclusions section for clarity. We truly appreciate your valuable feedback, which has helped enhance the depth and applicability of our work.
Once again, we extend our sincere gratitude for your time, effort, and thoughtful review. Your valuable input has greatly contributed to enhancing the quality of our work, and we truly appreciate your dedication to improving the research in this field.
Reviewer 2 Report
Comments and Suggestions for Authors
Dear Authors, you raised a good research idea a manuscript addresses critical research issue in the field of renewable energy, focusing on the automated detection and classification of soiling in PV modules.
1.Language and Grammar Issues: the manuscript contains many English grammatical problems, phrasing problems, and inconsistent terminology. Such as: "This comes at the expense of requiring more time and memory."
2. Lack of Technical Clarity: the manuscript should illustrate the hyperparameter settings, and data preprocessing steps clearly.
3.Inadequate Literature Review: missed recent advances in deep learning for PV systems and does not clearly articulate the research gap or novelty compared to existing studies.
4. lack of Data Description: How were images labeled? how data augmentation techniques applied? how you managed class imbalance issues?
5. Result and discussion sections: lack if comparison with Existing Methods with previous studies in PV soiling detection, low clarity of Confusion Matrix used.
6. Conclusion: avoid repetitive of results mentioned on abstract section. Generally, language and grammar, technical clarity, discussion of results with real world scenario may enhance the quality of this work.
Comments on the Quality of English LanguageIt needs English editing service.
Author Response
Reviewer 2
We sincerely appreciate the time and effort you have dedicated to reviewing our manuscript. Your thoughtful comments and constructive feedback have been invaluable in helping us enhance the clarity, technical rigor, and overall quality of our work. We are truly grateful for your insightful suggestions, which have guided us in refining our study and ensuring that it makes a meaningful contribution to the field of renewable energy.
Dear Authors, you raised a good research idea a manuscript addresses critical research issue in the field of renewable energy, focusing on the automated detection and classification of soiling in PV modules.
Response: We sincerely appreciate your kind words and thoughtful feedback on our manuscript. It is truly an honor to have our research recognized as a valuable contribution to the field of renewable energy. Your encouraging remarks and insightful comments have been invaluable in helping us refine and strengthen our work.
We have carefully reviewed and addressed each of your comments with great attention and consideration to ensure that our manuscript meets the highest standards of clarity, technical rigor, and relevance. Your constructive feedback has significantly contributed to improving the quality of our study, and we are truly grateful for your time and effort in reviewing our work.
Thank you once again for your time, consideration, and kind words.
- Language and Grammar Issues: the manuscript contains many English grammatical problems, phrasing problems, and inconsistent terminology. Such as: "This comes at the expense of requiring more time and memory."
Response: We sincerely appreciate the reviewer’s valuable feedback regarding the language and grammatical issues in our manuscript. To address these concerns, we have carefully revised the entire manuscript, correcting grammatical errors, improving phrasing, and ensuring consistency in terminology. Additionally, we have conducted a thorough language check to enhance the clarity and readability of our work.
We kindly request the reviewer to review the modifications, which have been highlighted in green for easy reference. We are grateful for your time and effort in assessing our work.
Thank you for your constructive comments.
- Lack of Technical Clarity: the manuscript should illustrate the hyperparameter settings, and data preprocessing steps clearly.
Response: We sincerely appreciate the reviewer’s valuable feedback and professional review regarding the need for greater clarity in explaining the hyperparameter settings and data preprocessing steps. To address this concern, we have carefully revised the relevant sections of the manuscript, providing detailed explanations of both aspects.
In Section 2.2 – Data Processing, we have expanded the description of the preprocessing pipeline applied to the dataset. The revised section now explicitly outlines the steps involved, including image resizing, color normalization, noise reduction using Gaussian filtering, histogram equalization for brightness correction, and data augmentation techniques such as rotation, brightness adjustment, and horizontal flipping. Additionally, we have included details on pixel value normalization, which ensures consistent feature scaling across images. These modifications enhance the clarity of our preprocessing methodology and demonstrate how these steps contribute to improving model robustness and classification accuracy.
Furthermore, in Section 3.4 – Experimental Configuration, we have reorganized and elaborated on the hyperparameter settings used during training. The revised section now presents a structured breakdown of key hyperparameters, including batch size, epochs, learning rate, momentum, weight decay, activation functions, and the optimizer used. Each parameter is accompanied by a brief explanation of its role in optimizing model performance. We have also provided justification for our choices, emphasizing how these values were determined through empirical testing to achieve a balance between computational efficiency and classification accuracy.
These modifications have been highlighted in blue in the revised manuscript for easy reference. We greatly appreciate the reviewer’s constructive suggestion, which has helped enhance the quality and readability of our manuscript.
Thank you for your time and insightful comments.
- Inadequate Literature Review: missed recent advances in deep learning for PV systems and does not clearly articulate the research gap or novelty compared to existing studies.
Response: We sincerely appreciate the reviewer’s valuable feedback regarding the literature review and the articulation of the research gap. In response to this comment, we have carefully revised and improved the manuscript as follows:
- We have updated the literature review to include recent studies and advancements in deep learning for PV systems, ensuring that our discussion reflects the latest developments in the field. Additionally, we have incorporated new references from recent research to strengthen the contextual background and provide a more comprehensive overview of existing methods. We kindly ask the reviewer to refer to the modifications highlighted in red in the Introduction for these updates.
- We have introduced a dedicated section that outlines the limitations of previous approaches and explains how our study addresses these gaps. Additionally, the contributions of our work are now more explicitly defined, ensuring a clearer distinction between our proposed method and existing techniques. We kindly ask the reviewer to refer to the newly added Subsection 1.1: Research Gaps and Contributions, where these improvements have been detailed.
We appreciate the reviewer’s insightful comments, which have helped us improve the clarity and depth of our work.
Thank you for your time and thoughtful feedback. We look forward to any further suggestions.
- lack of Data Description: How were images labeled? how data augmentation techniques applied? how you managed class imbalance issues?
Response: Thank you for your time, effort, and constructive comments. Your thoughtful feedback has been invaluable in improving the clarity and completeness of the manuscript. To enhance transparency and provide a more comprehensive explanation of the dataset preparation process, refinements have been made in Section 2.1: Data Collection and Section 2.2: Data Processing to further elaborate on image labeling, data augmentation techniques, and class imbalance handling. These updates ensure a clearer presentation of the applied techniques and strengthen the methodology. The revisions can be found in these sections, where the methodology has been further refined.
We sincerely appreciate your insightful contributions and the opportunity to improve this work. Please see the modifications highlighted in blue in the updated manuscript. Thank you once again for your time and effort, and we welcome any further suggestions you may have.
- Result and discussion sections: lack if comparison with Existing Methods with previous studies in PV soiling detection, low clarity of Confusion Matrix used.
Response: We sincerely appreciate the reviewer’s thoughtful and constructive feedback, as well as their professional and meticulous review, which have been invaluable in refining and enhancing the quality of our manuscript. Thank you for your time, effort, and dedication to providing such insightful guidance. In response to these insightful comments, we have made several significant enhancements to improve the clarity, depth, and contextualization of our findings. We have added Section 6: Discussion and Analysis, which provides a comprehensive comparative evaluation of our proposed approach against existing PV soiling detection methods. This addition highlights the advantages of our deep learning-based classification over traditional image-processing techniques and machine learning models, with a particular focus on the computational efficiency and accuracy of our Enhanced Artificial Neural Network (EANN) and Compact CNN models in comparison to conventional CNNs and other AI-based detection systems. Furthermore, we have expanded our discussion to include a comparison with sensor-based soiling detection approaches and threshold-based image-processing techniques, emphasizing the cost-effectiveness and real-time feasibility of our method for practical solar panel maintenance. Additionally, we have significantly improved the clarity of the confusion matrix representation by providing a detailed breakdown of classification results. We are truly grateful for the reviewer’s valuable insights, which have played a crucial role in strengthening our manuscript. We sincerely hope that these revisions adequately address the concerns raised and contribute to the clarity and comprehensiveness of our work. Thank you very much for your time, thoughtful review, and consideration.
- Conclusion: avoid repetitive of results mentioned on abstract section. Generally, language and grammar, technical clarity, discussion of results with real world scenario may enhance the quality of this work.
Response: We sincerely appreciate the reviewer’s valuable feedback and suggestions, which have helped us refine the manuscript. In response to these comments, we have carefully revised the Conclusion section to ensure that it does not repeat results already presented in the Abstract but instead provides a concise summary of the key findings, practical implications, and potential future directions. Please check the modified Conclusion section in the updated manuscript.
To further enhance technical clarity and readability, we have improved the language and grammar throughout the manuscript. Additionally, we have added a new section, 6. Discussion and Analysis, to provide a more detailed examination of our results in comparison with existing studies and their implications. Please check the new Section 6: Discussion and Analysis.
We are truly grateful for the reviewer’s insights, which have significantly contributed to the overall quality of our work. We hope these revisions adequately address the concerns raised. Thank you for your time, thoughtful review, and consideration.
Once again, we extend our sincere gratitude for your time, effort, and thoughtful review. Your valuable input has greatly contributed to enhancing the quality of our work, and we truly appreciate your dedication to improving the research in this field.
Round 2
Reviewer 1 Report
Comments and Suggestions for Authors
Thank you for the revisions,
Comments on the Quality of English LanguageThere are still some grammatical errors
Reviewer 2 Report
Comments and Suggestions for Authors
All my concerns and suggestions has been addressed!
Comments on the Quality of English LanguageIt has been addressed and modified.